# Energy-based Backdoor Defense Against Federated Graph Learning

**Guancheng Wan**[1†]   **Zitong Shi**[1†]   **Wenke Huang**[1†]   **Guibin Zhang**[3]
**Dacheng Tao**[4]   **Mang Ye**[1,2*]
[1] National Engineering Research Center for Multimedia Software,
School of Computer Science, Wuhan University
[2] Taikang Center for Life and Medical Sciences, Wuhan University
[3] Tongji University
[4] Generative AI Lab, College of Computing and Data Science, Nanyang Technological University

## Abstract

Federated Graph Learning is rapidly evolving as a privacy-preserving collaborative approach. However, backdoor attacks are increasingly undermining federated systems by injecting carefully designed triggers that lead the model making incorrect predictions. Trigger structures and injection locations in Federated Graph Learning are more diverse, making traditional federated defense methods less effective. In our work, we propose an effective **Fed**erated Graph Backdoor Defense using **T**opological **G**raph **E**nergy (**FedTGE**). At the client level, it injects distribution knowledge into the local model, assigning low energy to benign samples and high energy to the constructed malicious substitutes, and selects benign clients through clustering. At the server level, the energy elements uploaded by each client are treated as new nodes to construct a global energy graph for energy propagation, making the selected clients' energy elements more similar and further adjusting the aggregation weights. Our method can handle high data heterogeneity, does not require a validation dataset, and is effective under both small and large malicious proportions. Extensive results on various settings of federated graph scenarios under backdoor attacks validate the effectiveness of this approach. The code is available at https://github.com/ZitongShi/fedTGE.

## 1 Introduction

Federated Learning (FL) (Yang et al., 2019a; Mammen, 2021) has rapidly emerged as a significant research area in decentralized machine learning. This methodology allows multiple clients to collaboratively train a shared global model while preserving the privacy of sensitive data, thus eliminating the need to aggregate distributed data and ensuring adherence to privacy protocols (Zhang et al., 2021a; Kairouz et al., 2021). Consequently, FL presents a promising solution for training Graph Neural Networks (GNNs) on isolated graph data. Moreover, some existing work has utilized FL to train GNNs (Ju et al., 2024a; Kipf & Welling, 2016; Veličković et al., 2017), which we denote as Federated Graph Learning (FGL). While this distributed nature brings numerous benefits (Gilmer et al., 2017; Bruna et al., 2013), it also introduces additional vulnerabilities, particularly in the form of ***backdoor attacks*** from malicious participants (Chen et al., 2017; Li et al., 2022). These attacks involve injecting harmful data or models into the training process, embedding hidden behaviors that can trigger incorrect model outputs under specific conditions. These attacks aim to cause local models to learn incorrect information and activate the backdoor at critical times, resulting in erroneous predictions.

With the objective of better defending against these malicious attacks, defense methods against backdoor attacks in federated learning have been widely studied (Guerraoui et al., 2018; Yin et al., 2018a; Pillutla et al., 2022). Certain methods exclude outlier updates based on the statistical characteristics of model outputs. Some approaches examine pairwise distances among local models or the distances between local models and the global model to mitigate the influence of anomalous

---

*Corresponding author, † denotes equal contributions; each reserves the right to be listed first.

clients (Shejwalkar & Houmansadr, 2021). However, these methods often struggle to perform effectively in FGL environments, where graph data typically exhibit non-iid characteristics and complex topological structures. Some byzantine-robust federated learning methods require a clean and representative validation dataset (Cao et al., 2021). Consequently, they are less effective in scenarios where collecting a validation data set is challenging, such as in medical (Li et al., 2019b) and financial (Yang et al., 2019b) scenarios. Although recent studies (Huang et al., 2024b) have explored graph classification, there remains a significant gap in backdoor defense for node classification.

Based on the aforementioned discussion, we review the challenges existing in FGL under backdoor attacks. First of all, to address the high heterogeneity of the data, some methods choose to monitor the similarity of the updates of each client to adjust their contribution to the global model (Fung et al., 2018; Pillutla et al., 2022). However, the inherent topological complexity of graph data allows the trigger location and shape to be more arbitrary. It can be inserted at any position in the graph, leading to non-aligned updates, which hinders the effective identification of malicious clients and inevitably affects defense performance, as demonstrated in Table 1. Based on this observation, we raise the question: 1) *How can we design a backdoor defense method that can address scenarios where triggers exhibit complex topological characteristics?*

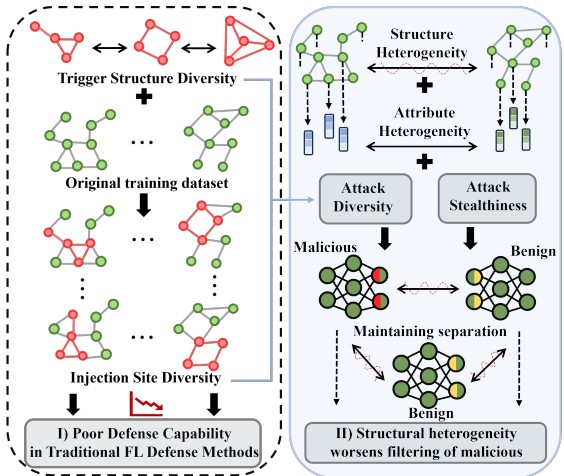

Figure 1: Problem illustration. We describe the challenges FGL encounters under backdoor attacks: **I)** Triggers vary in size, shape, and location of injection, making them more hidden. **II)** The structural heterogeneity introduced by FGL makes distinguishing between heterogeneous and malicious entities more difficult.

Secondly, some methods attempt to simply calculate the distances between clients or the similarity of certain distributions without any additional processing to differentiate between malicious and benign clients (Cao et al., 2021; Huang et al., 2023a), or filter out benign clients based on outlier detection (Shejwalkar & Houmansadr, 2021). However, these methods can easily misclassify perturbations caused by heterogeneity as malicious outliers due to their incomplete modeling of the data distribution. The additional structural heterogeneity introduced by FGL further complicates the ability to capture distributional information, inadvertently providing additional protection against backdoor attacks. This ultimately hinders the ability to effectively distinguish between malicious and benign clients in the metrics used for measurement, leading to the question: 2) *How can we learn structural distributions in a fine-grained manner and differentiate them from backdoor attacks to better filter out malicious clients?*

To address the two mentioned issues, we turn to energy-based models and explore their potential. Energy is an unnormalized probability likelihood (Song & Kingma, 2021), offering a flexible modeling approach that is not constrained by normalization. The strength of energy-based models lies in their ability to be integrated with virtually any model architecture. In our work, we combine energy with GCN to form an Energy-based GCN, preserving GCN's capability to capture complex structural information while benefiting from the flexibility of energy-based modeling. We introduce Topological Energy Client Clustering (TECC) to solve problem 1). TECC quantifies differences in client data distributions. Clients with significant energy distribution differences are marked as malicious and excluded from the aggregation process. We enhance local models by incorporating distribution awareness, combining their predictive capabilities with the ability to distinguish data energy distributions. Specifically, we add a final step in the training process to ensure the model assigns lower energy to the benign sample. However, indiscriminately lowering sample energy can lead to trivial solution. Therefore, we construct perturbed samples to simulate malicious triggers and raise the energy of these samples. Ultimately decoupling the distributions of benign and malicious samples. We then cluster the client energy distributions, identifying clients with significantly different energy distributions as potentially malicious.

To solve the second issue, further decoupling the energy distributions of malicious and benign clients, we propose Topological Energy Similarity Propagation (TESP). We collect the energy

distributions of each client and establish an energy graph based on the similarity of the energy distributions uploaded by the clients. Specifically, we consider the energy of samples uploaded by each client as its energy element. They are considered as new nodes for constructing the global energy graph. We then establish edges between highly similar energy elements to complete the construction of the energy graph. We enhance the similarity of energy distributions among clients that have established edge indices, thereby increasing the distinction between these clients and unselected malicious ones. Concurrently, energy elements with fewer established indices are considered *outliers* and are assigned lower transmission and aggregation weights. This energy adjustment in turn improves the clustering effectiveness of TECC. In synergy, this framework enables the model to learn effective topological distributions while achieving fine-grained decoupling of malicious and benign clients. We refer to the combination of these two strategies as **FedTGE**, an effective **Fed**erated Graph Backdoor Defense via **T**opological **G**raph **E**nergy. Our principal contributions are summarized as follows.

- We study a challenging problem: defending against backdoor attacks in Federated Graph Learning. Our focus is on mitigating these attacks while overcoming several assumptions made by existing methods, such as data homogeneity, the availability of validated samples, and the presence of a moderate proportion of malicious clients.
- We propose FedTGE, an innovative approach that addresses backdoor attacks characterized by complex topological triggers and highly arbitrary injection positions in FGL from the energy perspective. Our method enables clients to model the energy of graph structures at a fine-grained level, assigning higher aggregation weights to clients with high similarity in their energy distributions. This enhances the robustness of graph backdoor defenses.
- We conducted experiments on five mainstream datasets under both IID and Non-IID scenarios, as well as with varying proportions of malicious clients. The results demonstrate that our approach outperforms the current state-of-the-art methods in traditional FL.

## 2 RELATED WORK

### 2.1 FEDERATED GRAPH LEARNING

Federated Graph Learning (FGL) (Fu et al., 2022; Huang et al., 2023b; Li et al., 2023; Wan et al., 2024; Cai et al., 2024; Li et al., 2025; Fu et al., 2025) combines the characteristics of FL (Ye et al., 2023; Huang et al., 2024a; Liao et al., 2024) and GNNs (Chen et al., 2023b; Yin et al., 2024; Ju et al., 2024b), enabling collaborative learning of graph-structured data while preserving data privacy. In recent years, extensive research has focused on improving the generalization of the global model or obtaining personalized models that can span different graph domains (Wu et al., 2020; Chen et al., 2022; 2023a; 2024). However, the inherent heterogeneity and the complex, dynamic topology of graph data, along with the distributed nature of FGL, create significant vulnerabilities for backdoor attacks. Although there has been extensive work on effectively backdooring GNNs (Xi et al., 2021; Zhang et al., 2021c; Sun et al., 2020), there is a scarcity of research on backdoor defense paradigms specifically suited for FGL. To the best of our knowledge, we are the first to delve into backdoor defense in FGL, striving to create relevant benchmarks and contribute to this field.

### 2.2 BACKDOOR DEFENSE IN FEDERATED LEARNING

Malicious backdoor attackers pose a serious threat to federated systems (Huang et al., 2024b). To tackle the problem, researchers have proposed numerous defense methods, such as vector filtering techniques such as Bulyan (Guerraoui et al., 2018), RFA (Pillutla et al., 2022), and DnC (Shejwalkar & Houmansadr, 2021). Additionally, some defense methods utilize proxy data to further leverage server knowledge to defend against attacks, such as FLTrust (Cao et al., 2021) and Sageflow (Park et al., 2021). While most of these approaches can ensure successful defense under certain assumptions, they fail to provide stable defense performance in scenarios with non-iid data, difficulty in collecting proxy datasets, or a large number of attackers. Compared to traditional FL, FGL backdoor attacks often have triggers with higher randomness and greater stealth due to the more complex topological structure of graph data, making them more susceptible to attacks. While an excellent defense study has been proposed in FGL (Yang et al., 2024), it primarily focuses on graph classification tasks, and is not directly applicable to node classification tasks. We propose using energy as a bridge to address the aforementioned issues and fill the gap in backdoor defense for node classification.

## 2.3 ENERGY-BASED MODEL

The Energy-Based Model (EBM) is a generative model that directly models the unnormalized probability density function of the underlying data distribution. EBM represent the probability distribution of data by defining an energy function. A key feature of EBM is their flexibility: the energy function can be implemented using various forms of neural networks without strict structural constraints. This allows EBM to be adapted to a wide range of data types and tasks, including images, videos, and text (Deng et al., 2020; Arbel et al., 2020). In the domain of graphs, EBM have been applied to tasks such as substructure-preserving molecule design, molecular graph generation, and scene graph generation (Roy et al., 2023; Wu et al., 2023). The success of EBM in learning high-dimensional and complex molecular structures, such as proteins, underscores their powerful modeling capabilities (Cao & Shen, 2020; Xiao et al., 2023). In our work, we develop an Energy-Based GCN on the client side to model the energy of the entire graph. Benign samples that conform to the data distribution are assigned lower energy values. These energy distributions are subsequently uploaded, and on the server side, we further align the energy distributions among the selected clients. This alignment increases the separation between the energy elements of benign and malicious clients, thereby establishing a robust defense system against malicious attacks.

# 3 PRELIMINARY

## 3.1 FEDERATED GRAPH LEARNING

We follow the general paradigm of federated graph learning, where multiple clients collaboratively train a shared global model. Consider $K$ clients, indexed by $k$ and defined as $\mathcal{C} = \{c_k\}_{k=1}^K$. At the beginning of the $t^{th}$ communication round, we denote the current global model as $\mathcal{M}^t$ with parameters $w^t$, and the local model as $\mathcal{M}_k^t$ with corresponding parameters $w_k^t$. Each client $c_k$ possesses private data $\mathcal{G}_k = (\mathcal{V}_k, \mathcal{E}_k)$, where $\mathcal{V}_k = \{v_i\}_i^{N_k}$ represents the set of nodes containing $|\mathcal{V}_k| = N_k$ nodes, and $\mathcal{E}_k = \{e_{mn}\}_{m,n}$ denotes the set of edges. The adjacency matrix of $\mathcal{G}_k$ is defined as $\mathbf{A}_k = \{A_{ij}\}_{i,j}$, where $A_{ij} = 1$ if there is an edge between nodes $v_i$ and $v_j$, and $A_{ij} = 0$ otherwise. Similarly, $\mathbf{X}_k$ represents node features, and $\mathbf{Y}_k$ represents the corresponding label set.

## 3.2 ENERGY-BASED MODEL

Consider a sample $\mathbf{x} \in \mathbb{R}^D$. The energy-based model builds a function $E(\mathbf{x}, y) : \mathbb{R}^D \to \mathbb{R}$ that maps input instances with given labels to a scalar value, known as *energy*. The Boltzmann distribution expressed in terms of energy is represented as:

$$p(y|\mathbf{x}) = \frac{\exp(-E(\mathbf{x}, y))}{\sum_{y^*} \exp(-E(\mathbf{X}, y^*))}, \tag{1}$$

where $\sum_{y^*} \exp(-E(\mathbf{X}, y^*))$ is the partition function. Observe that in Eq. (1), it is very similar to our discriminative neural classifier. To relate the two, we set $E(\mathbf{x}, y) = -f(\mathbf{x})[y]$, where $f(\mathbf{x})[y]$ is the logits output of the model, and the energy function $E(x)$ can be formulated as follows:

$$E(x) = -\log \sum_{y^*} \exp(-E(\mathbf{x}, y^*)) = -\log \sum_y \exp(f(\mathbf{x})[y]). \tag{2}$$

# 4 METHODOLOGY

## 4.1 OVERVIEW

The overall framework of FedTGE is illustrated in Figure 2, and its algorithmic pseudocode is presented in Algorithm 1. At the client level, We inject structural energy awareness into the local models, lowering the energy of benign samples and raising that of malicious samples, respectively. At the server level, we cluster based on the differences in energy elements across clients to identify benign clusters. From a global perspective, we further construct an energy graph to enhance the similarity of the energy elements of the selected clients and adjust the aggregation weights accordingly.

## 4.2 TOPOLOGICAL ENERGY DIFFERENCE CLUSTERING

**Motivation.** In FGL backdoor attacks, the intricate topological structure of the graph data introduces considerable uncertainty in the methods used for trigger injection. This uncertainty is evident in both

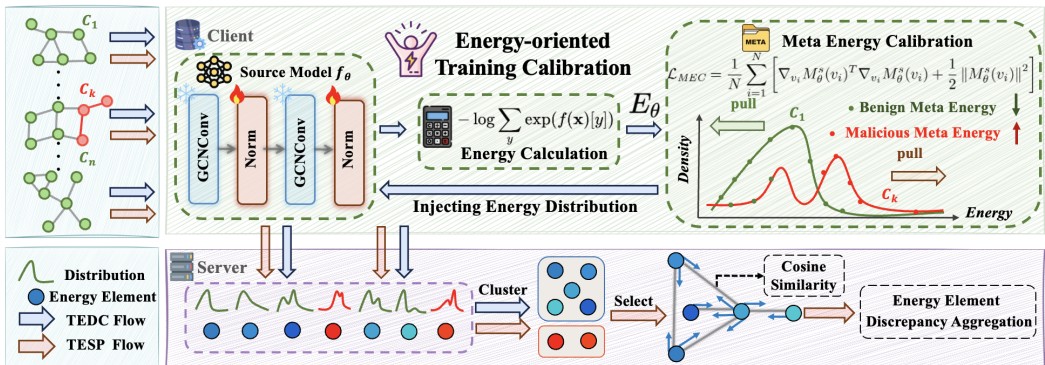

Figure 2: Architecture illustration of FedTGE. We used blue and red arrows to represent the two components of our method, TEDC and TESP, respectively. Best viewed in color. Zoom in for details.

the selection of injection positions and the diverse shapes that the triggers can assume. Although there are many effective defense paradigms in traditional FL, they often rely on impractical assumptions and are unable to handle triggers with complex topologies and diverse injection positions.

**Meta Energy.** We develop an Energy-Based GCN on the client side to model the energy of the entire graph, enhancing the model by injecting distributional knowledge of the samples. This enables the network to perform both node classification and distinguish the meta-energy of benign and malicious samples. First, we construct an energy-based model on top of the trained classifier:

$$E_\theta(x) = -\log \sum_y \exp(f_\theta(\mathbf{x})[y]). \tag{3}$$

For a node $v_i$, we define its output in the energy-based model simply as its meta energy: $M_e(v_i) = E_\theta(v_i)$. The meta energy represents the unnormalized likelihood of the sample point. Lower energy corresponds to higher likelihood and consequently a greater probability of the sample being benign. We then introduce the concept of perturbed meta energy, $\tilde{M}_e(\tilde{v}_i) = E_\theta(\tilde{v}_i^{adv})$, where $\tilde{v}_i^{adv}$ represents the perturbed version of $v_i$. Specifically, $\tilde{v}_i^{adv}$ is generated by arbitrarily adding or removing edges connected to $v_i$ and perturbing both its features and those of its neighbors. The objective is to inject a meta energy distribution into the original model by lowering $M_e(v_i)$ and raising $\tilde{M}_e(\tilde{v}_i^{adv})$. Let $d_i$ represent the degree of $v_i$, and $p$ denote the perturbation percentage. The generation of $\mathbf{X}^{adv}$ is achieved by perturbing the features of $\mathbf{X}$. Additionally, the adjacency matrix $\mathbf{A}^{adv}$ for $\tilde{v}_i^{adv}$ can be formulated as follows:

$$\begin{cases} A_{it}^{adv} = 1, t \neq i \text{ where } A_{it} = 0. \\ A_{ij}^{adv} = 0, j \neq i \text{ where } A_{ij} = 1. \end{cases} \tag{4}$$

**Meta Energy Calibration Objective.** The density function of the energy-based model is given by: $p_\theta(\mathbf{x}) = \exp(-E_\theta(\mathbf{x}))/Z_\theta$. Directly maximizing $p_\theta(v_i)$ to minimize $E_\theta(v_i)$ seems like a straightforward approach, but the normalization partition function $Z_\theta = \int \exp(-E_\theta(x))\,dx$ is typically very difficult to compute. Therefore, we consider using score matching to train the EBM.

Score matching is a technique for training EBMs by aligning the gradient of the log probability density function. By converting the distribution into its equivalent score, we can train EBMs more efficiently, as $\nabla_x \log p_\theta(x) = -\nabla_x E_\theta(x)$ eliminates the need for a normalization constant $Z_\theta$. However, traditional score matching only focuses on learning the data distribution and does not address the alignment of sample energy. This limitation, combined with the challenges posed by the highly discrete and topological nature of graph data, makes designing an effective score all the more critical.

**Definition 4.1. (Meta Energy Score):** *For $v_i$, we define the score assigned by the energy model as the meta energy score, which we use as a gradient surrogate in the discrete space:*

$$M_\theta^s(v_i) = \nabla E_\theta(v_i) = \left[ \frac{M_e(v_1) - \tilde{M}_e(\tilde{v}_1)}{M_e(v_1)}, \cdots, \frac{M_e(v_H) - \tilde{M}_e(\tilde{v}_H)}{M_e(v_H)} \right]. \tag{5}$$

In fact, Eq. (6) is equivalent to the following equation:

$$M_\theta^s(v_i) = \nabla E_\theta(v_i) = \left[ \frac{\log p_\theta(v_1) - \log p_\theta(\tilde{v}_1)}{\log p_\theta(v_1)}, \cdots, \frac{\log p_\theta(v_H) - \log p_\theta(\tilde{v}_H)}{\log p_\theta(v_H)} \right]. \tag{6}$$

Theoretically, the number of possible $\tilde{v}$ generated in this manner is infinite. We denote $H$ as the number of $\tilde{v}$ participating in the score calculation. This implies that using the gradient surrogate $\nabla E_\theta$ enables the model to learn the energy density distribution of the real data $p_{\text{data}}(v_i)$. With an effective score proxy for the gradient in place, we still follow the traditional score matching objective:

$$D_F(p_{\text{data}}(\mathbf{x}) \parallel p_\theta(\mathbf{x})) = \mathbb{E}_{p_{\text{data}}(\mathbf{x})}\left[\frac{1}{2}||\nabla_{\mathbf{x}} \log p_{\text{data}}(\mathbf{x}) - \nabla_{\mathbf{x}} \log p_\theta(\mathbf{x})||^2\right]. \tag{7}$$

With the energy score surrogate, our optimization objective is formulated as:

$$D_F(p_{\text{data}}(v_i) \parallel p_\theta(v_i)) = \mathbb{E}_{p_{\text{data}}(v_i)}\left[\frac{1}{2}||M_{\text{data}}^s(v_i) - M_\theta^s(v_i)||^2\right]. \tag{8}$$

However, since the $M_{\text{data}}^s(v_i)$ of the real data is unknown during the actual training of the model, we need to further optimize Eq. (8). Following (Hyvärinen & Dayan, 2005) and incorporating my gradient proxy while reducing computational complexity, we rewrite it as follows:

$$\mathcal{L}_{MEC} = \frac{1}{N}\sum_{i=1}^{N}\left[\nabla_{v_i}M_\theta^s(v_i)^T \nabla_{v_i}M_\theta^s(v_i) + \frac{1}{2}\|M_\theta^s(v_i)\|^2\right]. \tag{9}$$

In the loss function, we smooth the model output and minimize $\nabla_{v_i}E_\theta(v_i)$. As demonstrated in Definition 4.1, we effectively increase $\tilde{M}_e(\tilde{v}_i)/M_e(v_i)$. This aligns perfectly with our goal of incorporating knowledge of the data distribution into the model, allowing us to assign lower energy to benign samples and higher energy to malicious ones.

**Energy Element Discrepancy Cluster.** We calculate the meta energy for each sample of each client and refer to the collection of meta energy for each client as the energy element set, denoted as $\mathbf{E}_k$. We mark those energy elements that have significant differences from other clients and higher energy values as malicious clients and exclude them from the aggregation process. To systematically identify these anomalies, we use unsupervised FINCH clustering to filter out malicious clients. A comparison with popular clustering methods is provided in Table 3. As an example with three malicious clients, the pseudocode for the algorithm is shown in Algorithm 1.

---

**Algorithm 1** FedTGE

---

**Input:** *Communication rounds $T$, participant scale $K$, $k^{th}$ client private model $w_k$, and local data $\mathcal{G}_k$* **Output:** *The final global model $\mathcal{M}^T$*

**for** $t = 1, 2, \cdots, T$ **do**

    *Client Side:* **for** $k = 1$ *to* $K$ *in parallel* **do**

        $f_k(\cdot) \leftarrow \text{LocalUpdating}(w^t, \mathcal{G}_k)$ // Original training strategy

        $f_k^t(\cdot) \leftarrow \text{EnergyCalibrating}(f_k(\cdot), \mathcal{G}_k)$ // Injecting distribution knowledge

        $\mathbf{E}_k^t = \{E(f_k^t(v_i))\}_{i=1}^{N_k}$ // Calculating energy elements

    *Server Side:* // Cluster and find the cluster with the smaller mean

    $\{\mathbf{E}_a^t, \mathbf{E}_b^t\}$ and $\{\mathbf{E}_k^t\}_{k\neq a,b}^{\mathcal{N}}$ **where** $\text{mean}(\mathbf{E}_a^t, \mathbf{E}_b^t) \geq \text{mean}(\{\mathbf{E}_k^t\}_{k\neq a,b}^{\mathcal{N}})$

    $\mathbf{S}^t = \{\mathcal{S}_{mn}^t\}_{mn}^{\mathcal{N}} = \cos(\mathbf{E}_m^t, \mathbf{E}_n^t)_{m,n\neq a,b}$ // Calculating energy elements similarity

    $\mathcal{G}^e = (\mathcal{V}^e, \mathcal{E}^e) \leftarrow (\tau, \mathbf{S}^t)$ // Constructing energy graph

    $\mathbf{E}_k^{t*} \leftarrow (\beta_k^t, \{\mathbf{E}_k^t\}_{k\neq a,b}^{\mathcal{N}})$ // Energy graph similarity propagation

    $\mathcal{I}_k^t \leftarrow (\beta_k^t, \{\mathbf{E}_k^{t*}\}_{k\neq a,b}^{\mathcal{N}})$ // Energy disparity aggregation

    $w^{t+1} = \sum_{l=1}^{\mathcal{N}}\mathcal{I}_k w_k^t$ // Model parameter update

**return** $\mathcal{M}^T$

---

### 4.3 TOPOLOGICAL ENERGY SIMILARITY PROPAGATION

**Motivation.** Some defense methods simply measure certain distances between clients or certain distribution similarities without any additional processing to differentiate between malicious and benign clients. In scenarios with moderate to high proportions of malicious clients, these methods are susceptible to the combined effects of heterogeneity and backdoor attacks. This results in their inability to accurately filter out malicious clients, and in some cases, they even misclassify benign clients as malicious due to the heterogeneity.

**Construct Global Energy Graph.** Excluding the identified malicious clients, we utilize the energy elements of benign clients to compute the cosine similarity between each pair and construct a cosine similarity matrix, denoted as $\mathcal{S}$. We define the similarity between clients $c_m$ and $c_n$ as the element $\mathbf{S}_{mn}$ in the $m$-th row and $n$-th column of the matrix $\mathcal{S}$. Additionally, we set a threshold, denoted as $\tau$, to determine which samples are considered similar. When the value of $\mathbf{S}_{ij}$ is less than $\tau$, we consider the energy sequences of these two clients to be sufficiently similar. This implies that we can add an edge between these two clients in the global energy graph:

$$\mathbf{S} = [S_{mn}]_{m,n=1}^{\mathcal{N}}, \quad \text{where} \quad S_{mn} = \frac{\mathbf{E}_m \cdot \mathbf{E}_n}{\|\mathbf{E}_m\|\|\mathbf{E}_n\|}. \tag{10}$$

$$\mathcal{E}^e = [e_{mn}]_{m,n=1}^{\mathcal{N}}, \quad \text{where} \quad e_{mn} = \begin{cases} 1, & \text{if } \mathcal{S}_{mn} \geq \tau. \\ 0, & \text{if } \mathcal{S}_{mn} < \tau. \end{cases} \tag{11}$$

Here, $\mathcal{N}$ denotes the number of selected clients. The notation $\|\mathbf{E}_k\|$ represents the norm of the energy sequence for client $c_k$. $e_{mn}$ represents the edge between energy distributions $E_m$ and $E_n$, and $\tau$ is the set threshold. If $e_{mn} = 1$, it indicates that the two clients are sufficiently similar, and an edge will be established between them; otherwise, $e_{mn} = 0$.

**Energy Graph Similarity Propagation.** After establishing edge indices in the previous step, we obtain a global energy Graph with $\mathcal{N}$ nodes. From the above analysis, it is evident that $\mathbf{E}_k$ with more established indices has higher similarity with other clients. We consider these clients to be more benign and assign them higher propagation weights. We define energy transmission to occur over multiple rounds, and we consider the update rule for energy transmission as follows:

$$\mathbf{E}_k^* = \alpha \mathbf{E}_k \beta_k + \frac{(1-\alpha)}{n} \sum_{l=1}^{n} \mathbf{E}_k^l \beta_l, \quad \text{where} \quad \beta_k = \frac{d_k}{\sum_{l=1}^{\mathcal{N}} d_l}. \tag{12}$$

Here, $n$ represents the number of indices established by $\mathbf{E}_k$. $\mathbf{E}_k^l$ denotes the $l$-th neighbor of $\mathbf{E}_k$, and $\beta_k$ represents the energy propagation weight of $\mathbf{E}_k$.

**Energy Disparity Aggregation.** Conventional parameter aggregation treats all elements equally, failing to recognize their varying impacts on the target distribution. In our framework, we consider samples with lower energy to be more ***typical***. Commonly, low-energy samples are viewed as better fitting the model distribution, indicating they may be more suitable for training the model. Furthermore, a client with lower energy suggests a lower likelihood of malicious intent. Meanwhile, the more indices a client establishes, the higher the likelihood of it being benign. Therefore, we assign higher aggregation weights to such clients. This can be formalized as follows:

$$\mathcal{I}_k = \frac{\exp(-\mathbf{E}_k^*)}{\sum_{l \in \mathcal{N}} \exp(-\mathbf{E}_l^*)} \beta_k. \tag{13}$$

Here, $\mathcal{I}_k$ represents the aggregation weight assigned to client $k$.

$$w^{t+1} = \sum_{l=1}^{\mathcal{N}} \mathcal{I}_k w_k^t. \tag{14}$$

In this section, we consider two problematic scenarios: 1) If client $c_k$ shows insufficient similarity with certain clients $c_{m,n,l}$, we consider it a suspected misclassification ($c_k^{sus}$) and revoke its qualification to establish connections with $\tilde{\mathbf{E}}_k^{sus}$ and $\mathbf{E}_{m,n,l}$. 2) If a client $c_k$ has a similarity with all other clients below $\tau$, we consider $c_k$ a malicious client that has been mistakenly clustered with benign clients, revoking its right to participate in parameter aggregation.

**Discussion.** Energy-based models have been widely studied in various domains, including images (Song & Kingma, 2021), videos (LeCun et al., 2006), text (Deng et al., 2020), and graphs (Liu et al., 2021). The energy of a sample partially reflects the ***cost*** required for the model to learn that sample. The work Yuan et al. (2024) leverages the Energy-Based model to adapt to test-time samples. In contrast, our method filters out malicious client parameters and assigns higher aggregation weights to clients with lower energy. This results in higher purity of classification accuracy. Conversely, without filtering clients or employing conventional aggregation methods, backdoor samples may contribute to the classification accuracy, thereby falling into traps designed by attackers.

**Limitation.** At the micro level, our approach adjusts the energy of benign and malicious samples, while at the macro level, it performs unsupervised clustering of energy elements across different clients. Additionally, it further decouples benign and malicious energy elements and adjusts aggregation weights by constructing an energy graph from a global perspective. However, like many

Table 1: Comparison with **state-of-the-art** backdoor defense solutions in traditional federated learning over five mainstream datasets under both IID (upper) and Non-IID-Louvain (lower) settings. The best and second results are highlighted with bold and underline, respectively. Please see additional analysis in Section 5.2

| Methods | Cora | | | PubMed | | | Coauthor-CS | | | Coauthor-Phy | | | Amz-Photo | | |
|---|---|---|---|---|---|---|---|---|---|---|---|---|---|---|---|
| | $\mathcal{A}$ | $\mathcal{R}$ | $\mathcal{V}$ | $\mathcal{A}$ | $\mathcal{R}$ | $\mathcal{V}$ | $\mathcal{A}$ | $\mathcal{R}$ | $\mathcal{V}$ | $\mathcal{A}$ | $\mathcal{R}$ | $\mathcal{V}$ | $\mathcal{A}$ | $\mathcal{R}$ | $\mathcal{V}$ |
| *IID with a malicious proportion of $\Upsilon = 0.3$ and a trigger type of renyi* | | | | | | | | | | | | | | | |
| Vanilla | 74.25 | 19.47 | 46.86 | 86.12 | 4.26 | 45.19 | 85.90 | 20.47 | 53.19 | 93.51 | 16.37 | 54.94 | 81.61 | 5.33 | 43.47 |
| FLTrust | 73.75 | 60.00 | 66.88 | 83.45 | 31.30 | 57.38 | 86.22 | 57.76 | 71.99 | 94.19 | 40.72 | 67.46 | 82.82 | 83.07 | 82.95 |
| RSA | 72.42 | 46.40 | 59.41 | 85.42 | 6.16 | 46.29 | 85.75 | 13.92 | 52.34 | 93.93 | 22.61 | 58.27 | 84.79 | 23.81 | 53.30 |
| RLR | 76.00 | 13.07 | 44.53 | 86.77 | 14.7 8 | 50.78 | 84.02 | 15.37 | 49.70 | 93.56 | 14.53 | 54.05 | 78.26 | 6.84 | 42.55 |
| FLAME | 74.17 | 44.00 | 59.09 | 84.73 | 41.95 | 63.34 | 86.01 | 66.89 | 76.45 | 92.95 | 40.77 | 66.86 | 81.08 | 69.61 | 75.35 |
| G$^2$uard | 73.75 | 66.00 | 69.88 | 84.54 | 27.65 | 56.10 | 86.18 | 44.23 | 65.21 | 92.83 | 38.23 | 65.53 | 81.43 | 51.34 | 66.39 |
| Trim Median | 73.67 | 30.93 | 52.30 | 85.84 | 7.61 | 46.73 | 86.01 | 7.87 | 46.94 | 93.46 | 17.25 | 55.52 | 82.47 | 6.32 | 44.40 |
| Trimmed Mean | 73.17 | 25.60 | 49.38 | 86.08 | 5.72 | 45.90 | 86.11 | 6.05 | 46.09 | 93.35 | 17.55 | 55.45 | 81.74 | 6.32 | 44.03 |
| FreqFed | 76.25 | 18.67 | 47.56 | 86.16 | 7.25 | 46.71 | 86.93 | 10.93 | 48.93 | 93.38 | 6.78 | 50.08 | 81.76 | 4.33 | 43.05 |
| RFA | 78.42 | 40.27 | 59.34 | 87.12 | 11.28 | 49.20 | 85.49 | 9.07 | 47.28 | 93.35 | 14.47 | 53.91 | 84.26 | 7.27 | 45.77 |
| MMA | 75.00 | 36.27 | 55.63 | 86.98 | 6.67 | 46.83 | 87.04 | 13.77 | 50.41 | 93.87 | 22.16 | 58.02 | 85.55 | 23.81 | 54.68 |
| FoolsGold | 77.25 | 33.87 | 55.56 | 87.33 | 13.44 | 50.39 | 85.69 | 10.60 | 48.15 | 93.33 | 23.94 | 58.63 | 84.82 | 5.02 | 44.92 |
| DnC | 66.25 | 76.00 | 71.13 | 86.05 | 24.09 | 55.07 | 85.89 | 66.12 | 76.01 | 93.17 | 41.35 | 67.26 | 71.76 | 16.36 | 44.06 |
| FedCPA | 74.42 | 24.27 | 49.34 | 85.60 | 8.77 | 47.19 | 86.87 | 15.12 | 51.00 | 93.99 | 19.32 | 56.66 | 80.76 | 4.33 | 42.55 |
| Sageflow | 75.08 | 39.47 | 57.28 | 87.17 | 6.13 | 46.65 | 86.03 | 14.50 | 52.76 | 94.17 | 23.32 | 58.75 | 84.92 | 25.97 | 53.28 |
| FedTGE | 75.00 | 70.47 | **72.83** | 85.79 | 57.19 | 71.49 | 85.63 | 70.45 | **78.04** | 93.99 | 57.02 | **75.51** | 80.81 | 97.92 | **89.37** |

| Methods | Cora | | | PubMed | | | Coauthor-CS | | | Coauthor-Phy | | | Amz-Photo | | |
|---|---|---|---|---|---|---|---|---|---|---|---|---|---|---|---|
| | $\mathcal{A}$ | $\mathcal{R}$ | $\mathcal{V}$ | $\mathcal{A}$ | $\mathcal{R}$ | $\mathcal{V}$ | $\mathcal{A}$ | $\mathcal{R}$ | $\mathcal{V}$ | $\mathcal{A}$ | $\mathcal{R}$ | $\mathcal{V}$ | $\mathcal{A}$ | $\mathcal{R}$ | $\mathcal{V}$ |
| *Non-IID-Louvain with a malicious proportion of $\Upsilon = 0.3$ and a trigger type of renyi* | | | | | | | | | | | | | | | |
| Vanilla | 61.81 | 25.24 | 43.53 | 85.81 | 5.85 | 45.83 | 90.57 | 39.87 | 65.22 | 94.61 | 39.58 | 67.09 | 71.98 | 66.88 | 69.43 |
| FLTrust | 79.88 | 58.05 | 68.97 | 86.28 | 56.52 | 71.40 | 90.94 | 57.32 | 74.13 | 93.67 | 41.17 | 67.42 | 76.88 | 86.35 | 81.62 |
| RSA | 75.09 | 43.52 | 59.31 | 85.58 | 7.28 | 46.43 | 91.75 | 47.90 | 69.82 | 95.37 | 35.30 | 65.33 | 72.61 | 55.93 | 64.27 |
| RLR | 79.09 | 33.94 | 56.52 | 86.54 | 15.28 | 50.91 | 87.78 | 37.56 | 62.67 | 95.29 | 30.33 | 62.81 | 78.87 | 51.75 | 65.31 |
| FLAME | 75.05 | 55.23 | 65.14 | 84.56 | 52.78 | 68.67 | 87.49 | 66.23 | 76.86 | 92.23 | 57.76 | 75.00 | 69.65 | 88.78 | 79.22 |
| G$^2$uard | 73.66 | 47.82 | 60.74 | 83.34 | 33.47 | 58.41 | 88.15 | 55.89 | 72.02 | 93.38 | 41.87 | 67.63 | 72.34 | 67.29 | 69.82 |
| Trim Median | 63.85 | 22.15 | 43.00 | 86.01 | 17.85 | 51.93 | 87.68 | 50.84 | 69.26 | 69.26 | 45.96 | 70.24 | 63.45 | 76.13 | 69.79 |
| Trimmed Mean | 65.40 | 25.72 | 45.56 | 85.19 | 10.02 | 47.60 | 87.55 | 49.54 | 68.55 | 94.50 | 42.34 | 68.42 | 68.07 | 74.36 | 71.21 |
| FreqFed | 76.92 | 37.78 | 57.35 | 86.67 | 10.23 | 48.45 | 89.65 | 38.33 | 63.99 | 79.18 | 8.49 | 43.84 | 58.23 | 68.14 | 63.19 |
| RFA | 79.48 | 36.34 | 57.91 | 86.83 | 11.28 | 49.06 | 90.69 | 49.92 | 70.30 | 95.22 | 39.54 | 67.38 | 72.82 | 56.99 | 64.91 |
| MMA | 75.97 | 56.75 | 66.36 | 87.86 | 25.70 | 56.78 | 87.14 | 50.70 | 68.92 | 95.07 | 52.91 | 73.99 | 78.77 | 68.73 | 73.75 |
| FoolsGold | 76.86 | 38.24 | 57.55 | 86.77 | 13.27 | 50.02 | 90.96 | 43.09 | 67.02 | 95.34 | 35.22 | 65.28 | 84.99 | 51.23 | 68.11 |
| DnC | 41.21 | 80.05 | 60.63 | 60.40 | 13.12 | 36.76 | 53.21 | 77.43 | 65.32 | 85.48 | 64.69 | 75.08 | 45.90 | 94.09 | 69.99 |
| FedCPA | 77.60 | 40.14 | 58.87 | 86.70 | 22.36 | 54.53 | 89.31 | 43.14 | 66.23 | 95.31 | 35.45 | 65.38 | 77.36 | 54.69 | 66.03 |
| Sageflow | 79.80 | 54.38 | 67.09 | 87.88 | 23.47 | 55.67 | 89.42 | 53.86 | 71.64 | 93.26 | 53.36 | 73.31 | 77.10 | 64.75 | 70.93 |
| FedTGE | 77.32 | 55.85 | 66.58 | 86.79 | 67.22 | **77.01** | 88.15 | 72.10 | **80.13** | 94.06 | 57.98 | **76.02** | 77.46 | 94.81 | **86.14** |

popular solutions in FL, our method cannot effectively eliminate pre-existing poisoned parameters embedded in the model. Furthermore, our approach introduces some additional computational overhead due to the need to compute the energy distribution of samples, but it remains acceptable with a complexity of $\mathcal{O}(|N|)$.

## 5 EXPERIMENTS

We conducted experiments under both IID and Non-IID-Louvain (Wang et al., 2022; Zhang et al., 2021b) s2222ettings on five datasets to validate the superiority of our proposed FedTGE.

### 5.1 EXPERIMENTAL SETUP

**Datasets.** Adhering to (Liu et al., 2023), we evaluate the efficacy and robustness in three scenarios: Citation Network (Yang et al., 2016a), Co-authorship (Shchur et al., 2018), and Amz-purchase (McAuley et al., 2015). Detailed information about the datasets is provided in appendix A.

**Comparison Methods.** We compare FedTGE with several state-of-the-art methods in traditional FL: (1) FedAvg (McMahan et al., 2017b); (2) Trimmed Median and (3) Trimmed Mean (Yin et al., 2018b); (4) FoolsGold (Fung et al., 2018); (5) DnC (Shejwalkar & Houmansadr, 2021); (6) Sage-Flow (Park et al., 2021); (7) MMA (Huang et al., 2023a); (8) RFA (Pillutla et al., 2022); and (9)

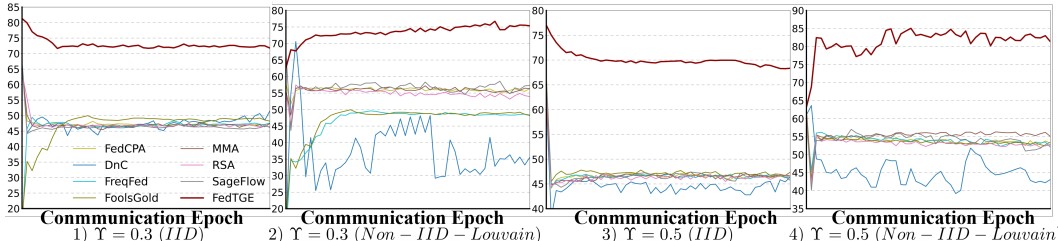

Figure 3: Comparison of **final metric** ($\mathcal{V}$) during the training process on Pubmed with $\Upsilon = 0.3$ and $0.5$ in both IID and Non-IID Louvain environments. Please see additional analysis in Section 5.2.

RLR (Ozdayi et al., 2021); (10) RSA (Li et al., 2019a); (11) Freqfed (Fereidooni et al., 2023); (12) FedCPA (Han et al., 2023); (13) FLAME (Nguyen et al., 2022); (14) FLTrust (Cao et al., 2021); (15) G$^2$uard (Yu et al., 2023). Detailed descriptions of these methods can be found in Appendix C.

**Network Structure.** Following the common approach in FGL(Dai et al., 2023), we utilize GCN as the 2 layers feature extractor and classifier, with the hidden layer size of 32 for all datasets.
**Backdoor Attack.** We demonstrate the effectiveness of the proposed method under the popular paradigm (Liu et al., 2023; Xu et al., 2021; Zheng et al., 2023). Considering the stealth of the backdoor attacks, The number of nodes in the trigger size is limited to 4 for all experiments, and its type and location is renyi and random, respectively. We set the malicious client ratio $\Upsilon$ as {0.1, 0.3, 0.5}, and conduct experiments under both IID and Non-IID-Louvain settings. Table 1 shows the results for $\Upsilon = 0.3$. Additional experimental results are provided in Appendix B, along with extensive energy distribution visualizations in Appendix E.

**Implement Details.** We provide the details from three views as:

- **Dataset Split**: In this paper, we conduct experiments on the node classification task. Following (Xu et al., 2021; Zheng et al., 2023), we partition the original training dataset (labeled) into training, validation, testing sets, comprising 60%, 20%, and 20% of the total nodes, respectively. Unlabeled nodes are leveraged for trigger injection and are subsequently relabeled with the target class.
- **Training Setting**: We repeat each experiment five times for each federated approaches to ensure the robustness and reliability of the results. The Adam optimizer (Kingma, 2014) with a learning rate of 0.01 is used to train the GNN models.
- **Evaluation Metric**: Following (Liu et al., 2023; Li et al., 2020; McMahan et al., 2017a), we use **node classification accuracy** ($\mathcal{A}$) and **backdoor failure rate** ($\mathcal{R}$) as our experimental metrics, as defined in Eq. (15). We define $\mathcal{V}$ as the final metric to evaluate the trade-off between accuracy and defense effectiveness.

$$\mathcal{A} = \frac{1}{N} \sum_{k=1}^{N} \frac{\sum_{\xi \in D_k^{\text{test}}} (\max(z) = y)}{|D_k^{\text{test}}|}, \quad \mathcal{R} = \frac{1}{\tilde{N}} \sum_{k=1}^{\tilde{N}} \left( 1 - \frac{\sum_{\tilde{\xi} \in \tilde{D}_k^{\text{test}}} (\max(\tilde{z}) = \tilde{y})}{|\tilde{D}_k^{\text{test}}|} \right), \quad \mathcal{V} = \frac{1}{2} (\mathcal{A} + \mathcal{R})$$

(15)

We define a query instance $\xi$ with logits output $z = \mathcal{M}^t(\xi)$, where $\mathcal{M}^t$ denotes the globally shared model. The mean accuracy across $N$ clients is denoted by $\mathcal{A}$. Similarly, backdoor queries $\tilde{\xi}$ and test samples $\tilde{D}_k^{\text{test}}$ yield a mean backdoor failure rate $\mathcal{R}$ across $\tilde{N}$ malicious clients. The final evaluation metric $\mathcal{V}$ is defined as the average of $\mathcal{A}$ and $\mathcal{R}$.

## 5.2 EXPERIMENTAL RESULTS

**Performance Comparison.** Table 1 shows the defense performance of conventional backdoor defense methods in traditional FL compared to our FedTGE approach under various settings. The results indicate that FedTGE consistently outperforms under both IID and Non-IID-Louvain conditions, demonstrating its effectiveness against diverse attack patterns in federated graph learning. Model Refinement Defenses such as RSA and RLR fail to detect the stealthy injection of triggers, rendering their defenses ineffective. Traditional defenses based on statistical distributions and standard distance metrics do not effectively learn structural distributions, leading to incorrect centroid calculations in heterogeneous environments or an inability to filter malicious entities due to metric coupling. In contrast, FedTGE maintained its defensive capabilities under these conditions.

**Convergence Analysis.** We plotted the curves of $\mathcal{V}$ during the training process on the Pubmed dataset with a malicious proportion of $\Upsilon = 0.3$ and $0.5$. The results are shown in Figure 3, where 1) and 2) represent the $\Upsilon = 0.3$ setting, and 3) and 4) represent the $\Upsilon = 0.5$ setting. We observe that FedTGE exhibits outstanding performance across all settings. Additionally, under high poisoning

rates, traditional FL methods essentially lose their defense capabilities, showing performance close to that of the standard FedAvg aggregation method.

## 5.3 DIAGNOSTIC ANALYSIS

**Key Components.** We conducted an ablation study on the key components of our method using the Cora and PubMed datasets, with a malicious client proportion of $\Upsilon = 30\%$. The results, demonstrating the effectiveness of each component, are presented in Table 2. TEDC significantly increases the backdoor failure rate $\mathcal{R}$, while TESP alleviates backdoor attacks to some extent on the server side through energy propagation and adjustment of aggregation weights.

**Different Cluster Methods.** We compared FINCH with several mainstream clustering strategies, including K-Means (Arthur & Vassilvitskii, 2006; Macqueen, 1967), DBSCAN (Ester et al., 1996), and OT (Cuturi, 2013; Solomon et al., 2015). The results are presented in Table 3. Both K-Means and DBSCAN require careful tuning of their respective hyper-parameters, which limits their effectiveness in heterogeneous federated environments. In contrast, the OT method is more sensitive to noise and has a higher computational complexity. On the other hand, FINCH does not require any hyper-parameter tuning and operates with near-linear complexity, making it more suitable for heterogeneous or federated systems with unknown scales.

Table 2: **Ablation on key components** for FedTGE on PubMed and Photo under Non-IID-Louvain. Please see details in Section 5.3

| TEDC Sec.5.3 | TESP Sec.5.3 | PubMed $\mathcal{A}$ | $\mathcal{R}$ | $\mathcal{V}$ | Photo $\mathcal{A}$ | $\mathcal{R}$ | $\mathcal{V}$ |
|---|---|---|---|---|---|---|---|
| ✗ | ✗ | 86.12 | 4.26 | 45.19 | 71.98 | 66.88 | 69.43 |
| ✓ | ✗ | 85.42 | 66.96 | 76.19 | 73.03 | 89.36 | 81.19 |
| ✗ | ✓ | 85.85 | 19.22 | 52.53 | 75.81 | 69.36 | 72.58 |
| ✓ | ✓ | 86.79 | 67.22 | **77.01** | 77.46 | 94.81 | **86.14** |

Table 3: **Ablation on popular clustering methods** for FedTGE on PubMed and Photo under Non-IID-Louvain. Please see details in Section 5.3

| Method | PubMed $\mathcal{A}$ | $\mathcal{R}$ | $\mathcal{V}$ | Photo $\mathcal{A}$ | $\mathcal{R}$ | $\mathcal{V}$ |
|---|---|---|---|---|---|---|
| K-Means | 87.32 | 60.19 | 73.75 | 72.44 | 91.50 | 81.97 |
| DBSCAN | 86.11 | 25.30 | 55.69 | 69.38 | 74.22 | 71.80 |
| OT | 69.34 | 13.99 | 41.66 | 79.57 | 71.34 | 75.45 |
| FINCH | 86.79 | 67.22 | **77.01** | 77.46 | 94.81 | **88.14** |

**Hyper-Parameters.** We performed a hyper-parameter ablation analysis on the Cora, PubMed, and Photo datasets. The analysis focused on key hyper-parameters: the number of energy calibration epochs at the client level (energy-epochs) and the threshold $\tau$ for edge index establishment at the server level. We observed that the choice of hyperparameters does not lead to significant fluctuations in the evaluation metrics. In most of our experiments, we set the default values as 10 for energy-epochs and 0.8 for $\tau$.

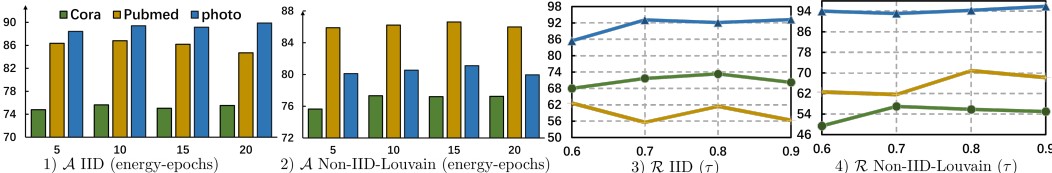

1) $\mathcal{A}$ IID (energy-epochs)    2) $\mathcal{A}$ Non-IID-Louvain (energy-epochs)    3) $\mathcal{R}$ IID ($\tau$)    4) $\mathcal{R}$ Non-IID-Louvain ($\tau$)

Figure 4: **Ablation Analysis** of Energy-Epochs and $\tau$ on the Cora, Pubmed, and Photo datasets under both IID and Non-IID settings with $\Upsilon = 0.3$. The results illustrate node classification accuracy $\mathcal{A}$ (left two panels) and backdoor failure rate $\mathcal{R}$ (right two panels).

## 6 CONCLUSION

In this paper, we are pioneers in innovatively exploring the problem of backdoor defenses in federated graph learning. We propose a novel framework called FedTGE, an effective **Fed**erated Graph Backdoor Defense via **T**opological **G**raph **E**nergy. At the client level, we inject the local models with energy-awareness, allowing them to learn the energy distribution of real data, assigning lower energy to benign samples and relatively higher energy to malicious samples. At the server level, We cluster clients based on their energy elements and determine their aggregation weights according to the similarity of energy elements between clients and the magnitude of each client's total energy. This method has demonstrated effectiveness and robustness across multiple scenarios. We hope this work offers a novel perspective for future research on federated graph backdoor defenses.

ACKNOWLEDGEMENT

This research is supported by the National Key Research and Development Project of China (2024YFC3308400), the National Natural Science Foundation of China (Grants 62361166629, 62176188, 623B2080), the Wuhan University Undergraduate Innovation Research Fund Project, the National Research Foundation, Singapore, and the CyberSG R&D Programme Office ("CRPO"), under the National Cybersecurity R&D Programme ("NCRP"), RIE2025 NCRP Funding Initiative (Award CRPO-GC1-NTU-002). The supercomputing system at the Supercomputing Center of Wuhan University supported the numerical calculations in this paper.

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

# A  DATASETS DETAILS

The statistics of the datasets used in our experiments are provided in Table 4.

- **Citation Network** (Cora, PubMed): The citation network datasets, such as Cora and PubMed, consist of interconnected research papers where nodes represent studies and edges denote citation relationships. They are often used in tasks such as classification of research papers and the construction of knowledge graphs, providing valuable information on research trends and hot topics within academic domains (Yang et al., 2016b).
- **Co-authorship** (CS, Physics): The co-authorship datasets, including those for fields such as Computer Science (CS) and Physics, are derived from the Microsoft Academic Graph. In these datasets, nodes represent authors, and edges signify co-author relationships. These datasets are used to predict the research fields of authors, aiding in the analysis of research collaboration networks, distribution of research areas, and academic influence (Shchur et al., 2018).
- **Amz-purchase** (Photo): The Amz-purchase datasets, such as the Photo dataset, are based on Amazon's co-purchase relations. In these datasets, nodes represent products, and edges indicate co-purchase relationships. The primary objective is to predict the category of each product, often used in recommendation systems and market analysis studies (McAuley et al., 2015).

# B  ADDITIONAL EXPERIMENT DETAILS

In this section, we conducted additional experiments to validate the superiority of FedTGE, covering different Non-IID settings (i.e., non-iid-louvain, non-iid-label-skew, non-iid-feature-skew), varying malicious ratios, and different trigger types. We introduce the key characteristics of these trigger types below:

- **Renyi Trigger** (Zhang et al., 2021c): Based on the Erdős–Rényi random graph model, the Renyi trigger introduces random nodes and edges into the graph. Each edge is created with an independent probability, resulting in a structure that lacks discernible patterns. This randomness offers strong obfuscation, making detection difficult.

- **GTA Trigger** (Xi et al., 2021): The GTA trigger employs well-designed, structured subgraphs (e.g., star or ring shapes) injected into the graph. By simultaneously modifying the features of the inserted nodes, the attacker amplifies their influence on targeted classifications. While this design leads to high attack success rates, its structural regularity increases the likelihood of detection by defense mechanisms.

- **WS Trigger** (Watts & Strogatz, 1998): Built on the Watts-Strogatz small-world model, the WS trigger introduces subgraphs characterized by high clustering coefficients and short average path lengths. These properties resemble real-world networks, enhancing their stealthiness. The high local clustering can significantly affect the graph's global properties, thereby improving attack efficacy. However, the attack success depends on carefully choosing parameters like rewiring probabilities.

- **BA Trigger** (Barabási & Albert, 1999): Derived from the Barabási-Albert scale-free network model, the BA trigger generates subgraphs with a power-law degree distribution, where a few "hub" nodes exhibit high connectivity. These hubs effectively propagate the backdoor effect across the graph, leveraging their high connectivity for stronger attacks. While this structure closely mimics real-world networks, the insertion of highly connected hub nodes might require extensive graph modifications, potentially impacting efficiency.

| Methods | Cora | | | PubMed | | | Coauthor-CS | | | Coauthor-Phy | | | Amz-Photo | | |
|---|---|---|---|---|---|---|---|---|---|---|---|---|---|---|---|
| | $\mathcal{A}$ | $\mathcal{R}$ | $\mathcal{V}$ | $\mathcal{A}$ | $\mathcal{R}$ | $\mathcal{V}$ | $\mathcal{A}$ | $\mathcal{R}$ | $\mathcal{V}$ | $\mathcal{A}$ | $\mathcal{R}$ | $\mathcal{V}$ | $\mathcal{A}$ | $\mathcal{R}$ | $\mathcal{V}$ |
| *IID with a malicious proportion of $\Upsilon = 0.5$ and a trigger type of renyi* | | | | | | | | | | | | | | | |
| Vanilla | 72.00 | 15.36 | 43.68 | 85.91 | 5.97 | 45.94 | 90.09 | 12.48 | 51.28 | 93.88 | 16.33 | 55.11 | 90.53 | 20.36 | 55.45 |
| FLTrust | 70.00 | 45.60 | 57.80 | 82.59 | 21.42 | 52.01 | 90.05 | 62.24 | 76.15 | 94.10 | 57.88 | 75.99 | 88.42 | 74.03 | 81.23 |
| RSA | 72.50 | 25.76 | 49.13 | 87.17 | 5.12 | 46.14 | 90.82 | 12.46 | 51.64 | 94.22 | 14.76 | 54.49 | 91.26 | 17.25 | 54.25 |
| RLR | 76.33 | 12.48 | 44.41 | 87.28 | 5.50 | 46.39 | 91.56 | 11.93 | 51.75 | 94.53 | 11.61 | 53.07 | 91.16 | 11.48 | 51.32 |
| FLAME | 70.92 | 28.00 | 49.46 | 84.61 | 24.77 | 54.69 | 90.41 | 55.36 | 72.89 | 93.45 | 56.06 | 74.76 | 87.76 | 76.66 | 82.21 |
| G$^2$uard | 69.17 | 45.60 | 57.38 | 83.47 | 28.58 | 56.03 | 88.74 | 65.79 | 77.27 | 93.61 | 52.89 | 73.26 | 89.92 | 64.56 | 77.24 |
| Trim Median | 69.17 | 11.68 | 40.42 | 85.58 | 5.02 | 45.30 | 90.37 | 9.36 | 49.86 | 93.46 | 11.85 | 52.65 | 89.71 | 18.29 | 54.00 |
| Trimmed Mean | 70.75 | 18.72 | 44.74 | 85.98 | 5.44 | 45.71 | 90.03 | 10.71 | 50.37 | 93.83 | 14.98 | 54.41 | 90.45 | 21.25 | 55.85 |
| FreqFed | 74.05 | 28.61 | 51.33 | 85.21 | 5.28 | 45.24 | 90.77 | 9.29 | 50.03 | 93.05 | 11.88 | 52.56 | 90.05 | 15.06 | 52.59 |
| RFA | 75.67 | 17.12 | 46.39 | 87.25 | 5.87 | 46.56 | 90.71 | 11.17 | 50.94 | 93.35 | 15.94 | 55.14 | 91.39 | 11.48 | 51.44 |
| MMA | 75.33 | 15.36 | 45.35 | 87.39 | 5.42 | 46.41 | 90.98 | 11.26 | 51.12 | 94.18 | 18.52 | 56.35 | 91.08 | 20.42 | 55.75 |
| FoolsGold | 73.83 | 15.52 | 44.68 | 87.17 | 5.56 | 46.37 | 90.87 | 11.89 | 51.38 | 94.19 | 15.01 | 54.60 | 91.13 | 10.13 | 50.63 |
| DnC | 62.33 | 55.04 | 58.69 | 83.77 | 4.65 | 44.21 | 87.61 | 82.19 | 84.75 | 92.65 | 57.45 | 75.05 | 87.29 | 20.68 | 53.98 |
| FedCPA | 74.17 | 11.20 | 42.68 | 86.01 | 5.18 | 45.59 | 90.60 | 10.93 | 50.77 | 93.37 | 13.22 | 53.30 | 90.05 | 9.61 | 49.83 |
| Sageflow | 75.58 | 26.24 | 50.91 | 87.14 | 5.28 | 46.21 | 90.55 | 5.46 | 48.01 | 92.99 | 17.33 | 55.16 | 91.18 | 17.25 | 54.22 |
| FedTGE | 72.80 | 73.76 | **73.28** | 85.47 | 53.56 | **69.52** | 90.21 | 80.63 | **85.42** | 93.69 | 62.52 | **78.10** | 90.55 | 92.62 | **91.59** |

| Methods | Cora | | | PubMed | | | Coauthor-CS | | | Coauthor-Phy | | | Amz-Photo | | |
|---|---|---|---|---|---|---|---|---|---|---|---|---|---|---|---|
| | $\mathcal{A}$ | $\mathcal{R}$ | $\mathcal{V}$ | $\mathcal{A}$ | $\mathcal{R}$ | $\mathcal{V}$ | $\mathcal{A}$ | $\mathcal{R}$ | $\mathcal{V}$ | $\mathcal{A}$ | $\mathcal{R}$ | $\mathcal{V}$ | $\mathcal{A}$ | $\mathcal{R}$ | $\mathcal{V}$ |
| *Non-IID-Louvain with a malicious proportion of $\Upsilon = 0.5$ and a trigger type of renyi* | | | | | | | | | | | | | | | |
| Vanilla | 60.13 | 22.37 | 41.25 | 85.44 | 12.21 | 48.82 | 89.75 | 36.69 | 63.22 | 92.24 | 33.85 | 63.05 | 66.16 | 52.89 | 59.53 |
| FLTrust | 78.07 | 50.87 | 64.47 | 84.44 | 44.71 | 64.58 | 90.29 | 62.24 | 76.27 | 93.27 | 65.93 | 79.60 | 71.12 | 75.08 | 73.10 |
| RSA | 72.91 | 24.83 | 48.87 | 87.67 | 16.44 | 52.05 | 90.91 | 48.38 | 69.64 | 94.36 | 40.64 | 68.00 | 74.26 | 35.44 | 54.85 |
| RLR | 80.38 | 28.85 | 54.62 | 87.68 | 18.75 | 53.21 | 89.60 | 42.96 | 66.28 | 95.24 | 36.17 | 65.70 | 61.02 | 30.99 | 46.00 |
| FLAME | 66.39 | 55.58 | 60.99 | 84.57 | 43.58 | 64.08 | 88.33 | 66.51 | 77.42 | 93.93 | 65.82 | 79.88 | 72.09 | 88.76 | 80.43 |
| G$^2$uard | 63.46 | 44.63 | 54.05 | 84.58 | 34.73 | 59.66 | 88.02 | 68.51 | 78.27 | 92.30 | 63.01 | 77.66 | 70.23 | 76.69 | 73.46 |
| Trim Median | 62.20 | 25.01 | 43.60 | 85.59 | 14.57 | 50.08 | 89.78 | 34.92 | 62.35 | 91.69 | 34.76 | 63.23 | 66.75 | 45.42 | 56.08 |
| Trimmed Mean | 62.26 | 29.26 | 45.78 | 85.89 | 14.46 | 50.18 | 89.46 | 33.69 | 61.57 | 93.55 | 57.33 | 75.44 | 66.62 | 49.24 | 57.93 |
| FreqFed | 73.05 | 28.61 | 50.83 | 85.63 | 18.55 | 53.09 | 90.84 | 40.36 | 65.60 | 94.31 | 38.82 | 66.57 | 73.30 | 29.45 | 51.38 |
| RFA | 78.14 | 38.78 | 58.46 | 87.63 | 16.28 | 51.95 | 90.81 | 45.15 | 67.98 | 95.41 | 44.55 | 69.98 | 81.29 | 45.29 | 63.29 |
| MMA | 74.68 | 37.41 | 56.04 | 86.93 | 22.91 | 54.92 | 88.15 | 41.43 | 64.79 | 94.07 | 50.68 | 72.37 | 76.82 | 38.60 | 57.71 |
| FoolsGold | 76.46 | 26.88 | 51.67 | 87.77 | 16.10 | 51.93 | 90.64 | 41.38 | 66.01 | 95.35 | 36.38 | 65.87 | 78.59 | 35.48 | 57.03 |
| DnC | 52.62 | 36.76 | 44.69 | 69.12 | 20.53 | 44.83 | 56.26 | 88.24 | 72.25 | 86.65 | 73.46 | 80.05 | 61.56 | 40.35 | 50.96 |
| FedCPA | 74.54 | 39.78 | 57.16 | 86.68 | 20.55 | 53.62 | 90.50 | 38.24 | 64.37 | 94.34 | 37.32 | 65.84 | 73.52 | 32.14 | 52.83 |
| Sageflow | 74.47 | 40.79 | 57.63 | 86.43 | 18.56 | 52.50 | 89.14 | 46.12 | 67.63 | 94.78 | 42.93 | 68.85 | 77.36 | 54.43 | 65.90 |
| FedTGE | 73.45 | 58.13 | **62.79** | 85.52 | 74.78 | **80.15** | 90.71 | 77.87 | **84.29** | 93.45 | 72.11 | **82.78** | 76.86 | 94.36 | **85.61** |

| Methods | Cora | | | PubMed | | | Coauthor-CS | | | Coauthor-Phy | | | Amz-Photo | | |
|---|---|---|---|---|---|---|---|---|---|---|---|---|---|---|---|
| | $\mathcal{A}$ | $\mathcal{R}$ | $\mathcal{V}$ | $\mathcal{A}$ | $\mathcal{R}$ | $\mathcal{V}$ | $\mathcal{A}$ | $\mathcal{R}$ | $\mathcal{V}$ | $\mathcal{A}$ | $\mathcal{R}$ | $\mathcal{V}$ | $\mathcal{A}$ | $\mathcal{R}$ | $\mathcal{V}$ |
| *IID with a malicious proportion of $\Upsilon = 0.1$ and a trigger type of renyi* | | | | | | | | | | | | | | | |
| Vanilla | 74.08 | 44.80 | 59.44 | 85.81 | 7.21 | 46.51 | 90.21 | 18.58 | 54.39 | 93.89 | 18.84 | 85.37 | 90.68 | 17.92 | 54.30 |
| FLTrust | 72.92 | 56.00 | 64.46 | 84.67 | 26.90 | 55.79 | 90.49 | 33.72 | 62.11 | 94.13 | 45.62 | 69.88 | 89.21 | 79.61 | 84.41 |
| RSA | 71.19 | 64.80 | 68.36 | 87.30 | 7.92 | 47.61 | 90.28 | 25.69 | 57.96 | 94.20 | 23.19 | 58.69 | 91.42 | 29.09 | 60.26 |
| RLR | 75.58 | 72.00 | 73.79 | 87.29 | 10.05 | 48.67 | 91.81 | 18.91 | 55.36 | 94.54 | 15.13 | 54.84 | 91.24 | 20.26 | 55.75 |
| FLAME | 73.91 | 64.00 | 68.96 | 85.12 | 34.31 | 59.72 | 89.83 | 39.07 | 64.45 | 92.27 | 47.82 | 70.05 | 90.56 | 64.42 | 77.49 |
| G$^2$uard | 74.17 | 48.00 | 61.09 | 84.69 | 34.87 | 59.78 | 89.63 | 33.46 | 61.55 | 93.44 | 34.69 | 64.07 | 89.34 | 79.61 | 84.48 |
| Trim Median | 69.33 | 53.60 | 61.47 | 85.63 | 6.60 | 46.12 | 90.05 | 19.45 | 54.75 | 93.71 | 21.16 | 57.43 | 90.26 | 20.26 | 55.26 |
| Trimmed Mean | 73.25 | 36.80 | 55.03 | 85.77 | 6.60 | 46.18 | 90.15 | 17.38 | 53.77 | 93.84 | 24.29 | 59.06 | 90.42 | 19.74 | 55.08 |
| FreqFed | 75.00 | 71.99 | 73.50 | 86.85 | 9.14 | 47.99 | 90.60 | 18.58 | 54.59 | 93.57 | 6.78 | 50.18 | 90.45 | 16.88 | 53.67 |
| RFA | 75.42 | 47.20 | 61.31 | 87.16 | 9.44 | 48.30 | 91.66 | 20.00 | 55.83 | 94.11 | 36.82 | 65.46 | 91.66 | 20.00 | 55.83 |
| MMA | 75.33 | 63.20 | 69.27 | 86.93 | 8.32 | 47.63 | 90.84 | 20.11 | 55.48 | 94.15 | 19.83 | 56.99 | 91.39 | 48.31 | 69.85 |
| FoolsGold | 76.17 | 44.00 | 60.08 | 87.42 | 8.63 | 48.03 | 90.89 | 19.13 | 55.01 | 94.27 | 21.10 | 57.69 | 91.39 | 16.62 | 54.01 |
| DnC | 64.50 | 80.80 | 72.65 | 84.38 | 8.12 | 46.25 | 83.44 | 36.97 | 60.21 | 92.67 | 46.78 | 69.73 | 88.26 | 9.61 | 48.94 |
| FedCPA | 74.17 | 64.00 | 49.08 | 85.29 | 9.14 | 47.22 | 89.98 | 20.77 | 55.37 | 93.20 | 24.41 | 58.80 | 90.66 | 24.68 | 57.67 |
| Sageflow | 76.08 | 60.80 | 68.44 | 87.04 | 9.54 | 48.29 | 90.09 | 22.62 | 56.36 | 94.14 | 25.51 | 59.82 | 91.50 | 47.01 | 69.26 |
| FedTGE | 75.42 | 84.00 | **79.71** | 85.12 | 46.19 | **65.66** | 90.31 | 40.43 | **65.37** | 93.28 | 57.10 | **70.42** | 90.76 | 98.44 | **94.60** |

| Methods | Cora | | | PubMed | | | Coauthor-CS | | | Coauthor-Phy | | | Amz-Photo | | |
|---|---|---|---|---|---|---|---|---|---|---|---|---|---|---|---|
| | $\mathcal{A}$ | $\mathcal{R}$ | $\mathcal{V}$ | $\mathcal{A}$ | $\mathcal{R}$ | $\mathcal{V}$ | $\mathcal{A}$ | $\mathcal{R}$ | $\mathcal{V}$ | $\mathcal{A}$ | $\mathcal{R}$ | $\mathcal{V}$ | $\mathcal{A}$ | $\mathcal{R}$ | $\mathcal{V}$ |
| *Non-IID-Louvain with a malicious proportion of $\Upsilon = 0.1$ and a trigger type of renyi* | | | | | | | | | | | | | | | |
| Vanilla | 62.52 | 40.69 | 51.60 | 85.41 | 22.19 | 53.80 | 90.08 | 35.30 | 62.69 | 92.54 | 46.91 | 69.72 | 74.44 | 70.79 | 72.61 |
| FLTust | 78.69 | 68.00 | 73.35 | 85.63 | 84.69 | 85.16 | 91.14 | 69.51 | 80.33 | 93.91 | 62.00 | 77.96 | 77.81 | 72.69 | 75.25 |
| RSA | 74.39 | 78.68 | 76.54 | 88.05 | 21.79 | 54.92 | 91.79 | 55.16 | 73.47 | 95.36 | 55.16 | 75.26 | 74.39 | 78.68 | 76.54 |
| RLR | 80.46 | 65.52 | 72.99 | 87.95 | 24.18 | 56.07 | 90.04 | 41.71 | 65.87 | 95.52 | 44.45 | 69.99 | 65.85 | 63.16 | 65.51 |
| FLAME | 75.00 | 68.89 | 71.95 | 85.53 | 82.78 | 84.16 | 89.69 | 69.51 | 79.60 | 93.74 | 61.73 | 77.74 | 73.29 | 70.28 | 71.79 |
| G$^2$uard | 70.49 | 66.00 | 68.25 | 85.54 | 83.73 | 84.64 | 88.89 | 65.13 | 77.01 | 93.06 | 58.98 | 76.02 | 73.12 | 73.56 | 73.34 |
| Trim Median | 65.07 | 46.90 | 55.98 | 86.23 | 21.09 | 53.66 | 90.15 | 41.21 | 65.68 | 93.29 | 53.01 | 73.15 | 66.36 | 65.26 | 65.81 |
| Trimmed Mean | 65.12 | 44.83 | 54.97 | 86.20 | 20.50 | 53.35 | 90.18 | 38.36 | 64.27 | 93.85 | 50.20 | 72.02 | 69.70 | 67.63 | 68.67 |
| FreqFed | 77.71 | 55.17 | 66.44 | 86.63 | 25.37 | 56.50 | 77.39 | 61.84 | 69.61 | 78.42 | 40.47 | 59.44 | 75.39 | 61.84 | 68.61 |
| RFA | 79.41 | 76.55 | 77.98 | 87.88 | 24.98 | 56.43 | 91.14 | 48.19 | 69.66 | 95.38 | 51.68 | 73.53 | 80.74 | 67.37 | 74.05 |
| MMA | 77.49 | 60.26 | 68.88 | 87.60 | 25.27 | 56.44 | 86.72 | 66.62 | 76.66 | 94.88 | 57.42 | 76.15 | 80.20 | 60.26 | 70.23 |
| FoolsGold | 78.84 | 62.07 | 70.45 | 87.86 | 25.57 | 56.77 | 91.20 | 46.90 | 69.05 | 95.44 | 51.09 | 73.26 | 85.24 | 62.89 | 74.07 |
| DnC | 41.94 | 1.00 | 70.97 | 57.57 | 90.95 | 74.56 | 56.50 | 67.40 | 61.95 | 86.01 | 71.13 | 78.57 | 44.06 | 95.53 | 69.80 |
| FedCPA | 77.43 | 78.76 | **78.09** | 86.68 | 16.55 | 51.62 | 90.01 | 48.54 | 69.90 | 94.40 | 56.64 | 75.52 | 75.53 | 71.05 | 73.29 |
| Sageflow | 76.13 | 60.53 | 68.33 | 87.76 | 26.77 | 57.26 | 86.30 | 61.28 | 73.79 | 95.14 | 55.00 | 75.07 | 76.13 | 66.84 | 71.49 |
| FedTGE | 78.76 | 73.68 | 76.22 | 86.54 | 95.52 | **91.03** | 89.23 | 77.37 | **83.30** | 94.66 | 69.75 | **82.21** | 75.50 | 80.26 | **77.89** |

| Methods | Cora $\mathcal{A}$ | $\mathcal{R}$ | $\mathcal{V}$ | PubMed $\mathcal{A}$ | $\mathcal{R}$ | $\mathcal{V}$ | Coauthor-CS $\mathcal{A}$ | $\mathcal{R}$ | $\mathcal{V}$ | Coauthor-Phy $\mathcal{A}$ | $\mathcal{R}$ | $\mathcal{V}$ | Amz-Photo $\mathcal{A}$ | $\mathcal{R}$ | $\mathcal{V}$ |
|---|---|---|---|---|---|---|---|---|---|---|---|---|---|---|---|
| *Non-IID-Label-Skew with $\alpha = 0.5$ and a malicious proportion of $\Upsilon = 0.3$ under a trigger type of renyi.* | | | | | | | | | | | | | | | |
| Vanilla | 61.72 | 73.89 | 67.80 | 85.88 | 11.28 | 48.58 | 87.69 | 59.69 | 73.69 | 94.34 | 39.54 | 66.94 | 77.39 | 54.46 | 69.93 |
| FLTrust | 63.12 | 59.66 | 61.39 | 82.63 | 8.98 | 45.80 | 91.96 | 64.92 | 78.44 | 94.58 | 31.25 | 62.92 | 77.55 | 72.06 | 74.81 |
| RSA | 67.51 | 50.15 | 58.83 | 85.17 | 12.03 | 48.60 | 91.67 | 62.98 | 77.33 | 94.32 | 54.61 | 74.47 | 78.41 | 56.35 | 67.38 |
| RLR | 65.62 | 51.04 | 58.33 | 85.33 | 0.64 | 42.99 | 92.03 | 58.65 | 75.34 | 94.67 | 46.85 | 70.76 | 78.93 | 64.51 | 71.72 |
| FLAME | 63.27 | 70.08 | 66.68 | 84.28 | 14.43 | 49.36 | 86.08 | 66.29 | 76.19 | 90.27 | 42.64 | 66.46 | 77.78 | 51.78 | 64.78 |
| G$^2$uard | 63.37 | 63.56 | 63.47 | 83.16 | 7.78 | 45.47 | 85.90 | 68.92 | 77.41 | 91.27 | 37.23 | 64.25 | 77.89 | 58.12 | 68.01 |
| Trim Median | 60.90 | 58.35 | 59.62 | 86.39 | 7.91 | 47.15 | 87.49 | 51.69 | 69.59 | 94.24 | 30.72 | 62.48 | 77.28 | 58.66 | 67.97 |
| Trimmed Mean | 54.99 | 66.89 | 60.94 | 85.73 | 2.13 | 43.93 | 88.49 | 44.59 | 66.54 | 94.16 | 33.82 | 63.99 | 78.25 | 58.64 | 68.45 |
| FreqFed | 67.11 | 50.13 | 58.62 | 86.63 | 10.82 | 48.73 | 91.74 | 63.40 | 77.57 | 94.64 | 47.53 | 71.09 | 77.87 | 55.19 | 66.53 |
| RFA | 65.23 | 74.39 | 69.81 | 86.03 | 0.30 | 43.16 | 91.10 | 58.93 | 75.02 | 94.84 | 58.90 | 76.87 | 76.22 | 56.32 | 66.27 |
| MMA | 65.67 | 62.24 | 63.96 | 87.48 | 10.44 | 48.96 | 91.80 | 38.70 | 65.75 | 94.80 | 28.70 | 61.75 | 76.33 | 52.04 | 64.19 |
| FoolsGold | 65.22 | 63.98 | 64.60 | 86.86 | 10.91 | 48.88 | 91.31 | 61.05 | 76.18 | 91.31 | 61.05 | 77.18 | 77.88 | 62.73 | 70.31 |
| DnC | 54.14 | 87.58 | 70.86 | 69.43 | 18.11 | 43.77 | 78.79 | 61.66 | 70.23 | 90.05 | 31.81 | 60.93 | 68.36 | 62.64 | 65.50 |
| FedCPA | 66.41 | 54.43 | 60.42 | 85.88 | 11.39 | 48.64 | 91.48 | 57.24 | 74.36 | 94.68 | 36.41 | 65.54 | 78.01 | 53.74 | 65.88 |
| Sageflow | 65.42 | 44.91 | 55.17 | 86.09 | 0.35 | 43.22 | 91.57 | 64.46 | 78.02 | 94.59 | 26.51 | 60.55 | 78.25 | 50.51 | 64.38 |
| FedTGE | 64.28 | 88.91 | **76.60** | 86.98 | 43.65 | **65.32** | 91.68 | 77.96 | **84.82** | 94.79 | 86.79 | **90.79** | 79.65 | 92.65 | **86.15** |

| Methods | Cora $\mathcal{A}$ | $\mathcal{R}$ | $\mathcal{V}$ | PubMed $\mathcal{A}$ | $\mathcal{R}$ | $\mathcal{V}$ | Coauthor-CS $\mathcal{A}$ | $\mathcal{R}$ | $\mathcal{V}$ | Coauthor-Phy $\mathcal{A}$ | $\mathcal{R}$ | $\mathcal{V}$ | Amz-Photo $\mathcal{A}$ | $\mathcal{R}$ | $\mathcal{V}$ |
|---|---|---|---|---|---|---|---|---|---|---|---|---|---|---|---|
| *Non-IID-Feature-Skew with $\alpha = 0.5$ and a malicious proportion of $\Upsilon = 0.3$ under a trigger type of renyi.* | | | | | | | | | | | | | | | |
| Vanilla | 62.97 | 44.79 | 53.88 | 84.00 | 31.60 | 57.80 | 88.97 | 59.52 | 74.23 | 94.05 | 48.23 | 71.14 | 77.46 | 58.43 | 67.95 |
| FLTrust | 60.47 | 55.56 | 58.02 | 83.68 | 32.17 | 57.93 | 89.09 | 67.97 | 78.52 | 94.55 | 55.36 | 74.96 | 78.69 | 69.05 | 73.87 |
| RSA | 63.10 | 31.54 | 47.32 | 86.79 | 31.91 | 59.35 | 90.49 | 60.30 | 75.40 | 93.79 | 49.23 | 71.51 | 80.78 | 62.75 | 71.77 |
| RLR | 65.12 | 33.38 | 49.25 | 87.46 | 19.61 | 53.54 | 91.01 | 57.94 | 74.48 | 94.88 | 46.23 | 70.56 | 80.14 | 66.45 | 73.30 |
| FLAME | 60.30 | 66.50 | 63.40 | 83.48 | 62.34 | 72.91 | 87.24 | 69.98 | 78.61 | 92.85 | 62.32 | 77.59 | 78.52 | 74.65 | 76.59 |
| G$^2$uard | 59.82 | 74.19 | 67.00 | 82.66 | 52.65 | 67.66 | 86.47 | 68.67 | 77.57 | 92.88 | 60.11 | 76.50 | 79.54 | 64.60 | 72.70 |
| Trim Median | 62.97 | 57.44 | 60.21 | 84.10 | 34.23 | 59.17 | 88.60 | 54.11 | 71.16 | 93.89 | 55.23 | 74.56 | 78.31 | 65.27 | 71.79 |
| Trimmed Mean | 64.78 | 54.10 | 59.44 | 84.53 | 44.23 | 64.38 | 88.34 | 60.47 | 74.41 | 93.23 | 52.36 | 72.80 | 78.98 | 57.19 | 68.09 |
| FreqFed | 66.39 | 47.99 | 57.19 | 85.12 | 24.71 | 54.92 | 90.13 | 65.25 | 77.69 | 94.29 | 56.69 | 75.49 | 90.76 | 53.37 | 72.07 |
| RFA | 65.23 | 55.98 | 60.61 | 86.89 | 43.41 | 65.15 | 89.15 | 62.93 | 76.04 | 94.56 | 34.25 | 65.41 | 80.57 | 63.37 | 71.97 |
| MMA | 66.67 | 60.81 | 63.74 | 87.42 | 51.09 | 69.26 | 90.65 | 58.79 | 74.72 | 93.46 | 28.56 | 61.01 | 81.56 | 64.60 | 73.08 |
| FoolsGold | 66.38 | 43.21 | 54.79 | 86.10 | 35.35 | 60.73 | 89.77 | 57.14 | 73.46 | 94.02 | 44.58 | 69.30 | 80.34 | 43.37 | 61.86 |
| DnC | 57.53 | 67.61 | 62.57 | 84.60 | 36.19 | 60.39 | 81.62 | 48.90 | 65.26 | 88.71 | 73.21 | 80.96 | 72.11 | 73.98 | 73.05 |
| FedCPA | 68.07 | 55.77 | 61.92 | 87.76 | 54.35 | 71.06 | 90.91 | 63.26 | 77.09 | 94.94 | 58.69 | 76.82 | 79.89 | 64.61 | 72.25 |
| Sageflow | 65.98 | 56.37 | 61.18 | 87.99 | 60.24 | 74.12 | 89.64 | 72.96 | 81.30 | 93.81 | 69.98 | 81.90 | 80.67 | 65.84 | 73.26 |
| FedTGE | 63.69 | 88.98 | **76.34** | 86.53 | 71.29 | **78.91** | 90.23 | 79.96 | **85.10** | 94.40 | 85.70 | **90.05** | 79.63 | 92.34 | **85.99** |

| Methods | Cora | | | PubMed | | | Coauthor-CS | | | Coauthor-Phy | | | Amz-Photo | | |
|---|---|---|---|---|---|---|---|---|---|---|---|---|---|---|---|
| | $\mathcal{A}$ | $\mathcal{R}$ | $\mathcal{V}$ | $\mathcal{A}$ | $\mathcal{R}$ | $\mathcal{V}$ | $\mathcal{A}$ | $\mathcal{R}$ | $\mathcal{V}$ | $\mathcal{A}$ | $\mathcal{R}$ | $\mathcal{V}$ | $\mathcal{A}$ | $\mathcal{R}$ | $\mathcal{V}$ |

*Non-IID-Louvain with a malicious proportion of $\Upsilon = 0.3$ under a trigger type of gta.*

| Methods | Cora $\mathcal{A}$ | $\mathcal{R}$ | $\mathcal{V}$ | PubMed $\mathcal{A}$ | $\mathcal{R}$ | $\mathcal{V}$ | CS $\mathcal{A}$ | $\mathcal{R}$ | $\mathcal{V}$ | Phy $\mathcal{A}$ | $\mathcal{R}$ | $\mathcal{V}$ | Amz $\mathcal{A}$ | $\mathcal{R}$ | $\mathcal{V}$ |
|---|---|---|---|---|---|---|---|---|---|---|---|---|---|---|---|
| Vanilla | 78.54 | 27.96 | 53.25 | 86.86 | 11.70 | 49.28 | 83.92 | 54.82 | 69.37 | 93.42 | 24.86 | 59.14 | 72.32 | 50.32 | 61.32 |
| FLTrust | 78.21 | 43.01 | 60.61 | 86.36 | 36.05 | 61.21 | 91.23 | 79.26 | 85.25 | 94.21 | 59.26 | 76.74 | 74.12 | 75.30 | 74.71 |
| RSA | 84.52 | 18.79 | 51.65 | 89.15 | 16.55 | 52.85 | 91.06 | 47.18 | 69.12 | 94.61 | 42.93 | 68.77 | 73.62 | 35.59 | 54.61 |
| RLR | 83.65 | 27.48 | 55.56 | 88.88 | 17.20 | 53.04 | 91.28 | 75.42 | 83.35 | 94.99 | 41.56 | 68.27 | 72.83 | 50.45 | 61.64 |
| FLAME | 77.54 | 67.13 | 72.34 | 85.45 | 40.51 | 62.98 | 89.62 | 78.32 | 83.97 | 93.28 | 50.49 | 71.89 | 73.99 | 92.69 | 83.34 |
| G²uard | 75.31 | 69.62 | 72.47 | 86.12 | 42.90 | 64.51 | 91.31 | 73.46 | 82.39 | 93.22 | 61.54 | 77.38 | 72.14 | 83.83 | 77.99 |
| Trim Median | 79.49 | 33.06 | 56.26 | 86.63 | 14.58 | 50.60 | 84.16 | 59.94 | 72.05 | 93.55 | 27.39 | 60.47 | 73.76 | 59.91 | 66.83 |
| Trimmed Mean | 76.50 | 23.61 | 50.05 | 87.23 | 18.43 | 52.83 | 84.37 | 59.03 | 71.69 | 93.91 | 23.26 | 58.59 | 75.25 | 51.64 | 63.44 |
| FreqFed | 81.91 | 29.62 | 55.76 | 87.86 | 15.60 | 51.73 | 90.49 | 55.92 | 73.21 | 94.95 | 40.96 | 67.95 | 73.20 | 48.89 | 61.05 |
| RFA | 82.06 | 29.39 | 55.73 | 88.87 | 16.23 | 52.55 | 90.16 | 62.78 | 76.47 | 94.50 | 45.13 | 69.81 | 73.36 | 43.67 | 58.52 |
| MMA | 79.72 | 46.95 | 63.34 | 88.11 | 23.91 | 56.01 | 90.43 | 75.92 | 83.17 | 94.63 | 39.62 | 67.12 | 71.49 | 58.13 | 64.81 |
| FoolsGold | 79.56 | 25.60 | 52.58 | 89.15 | 16.07 | 52.61 | 90.78 | 48.48 | 69.63 | 94.68 | 40.56 | 67.62 | 72.97 | 43.39 | 58.18 |
| DnC | 67.90 | 56.82 | 62.36 | 75.92 | 32.83 | 54.37 | 58.25 | 90.42 | 74.33 | 79.51 | 58.64 | 69.07 | 62.79 | 95.08 | 78.93 |
| FedCPA | 78.79 | 24.00 | 51.40 | 88.42 | 14.62 | 51.52 | 90.64 | 60.24 | 75.44 | 94.55 | 40.16 | 67.35 | 72.97 | 38.36 | 55.67 |
| Sageflow | 79.95 | 29.49 | 54.72 | 89.04 | 16.70 | 52.87 | 90.09 | 70.80 | 80.45 | 94.54 | 54.69 | 74.62 | 73.49 | 42.51 | 57.99 |
| FedTGE | 80.23 | 75.23 | **77.73** | 87.56 | 55.34 | **71.45** | 90.53 | 87.82 | **89.18** | 94.39 | 72.37 | **83.38** | 74.46 | 97.56 | **86.01** |

| Methods | Cora | | | PubMed | | | Coauthor-CS | | | Coauthor-Phy | | | Amz-Photo | | |
|---|---|---|---|---|---|---|---|---|---|---|---|---|---|---|---|
| | $\mathcal{A}$ | $\mathcal{R}$ | $\mathcal{V}$ | $\mathcal{A}$ | $\mathcal{R}$ | $\mathcal{V}$ | $\mathcal{A}$ | $\mathcal{R}$ | $\mathcal{V}$ | $\mathcal{A}$ | $\mathcal{R}$ | $\mathcal{V}$ | $\mathcal{A}$ | $\mathcal{R}$ | $\mathcal{V}$ |

*Non-IID-Louvain with a malicious proportion of $\Upsilon = 0.3$ under a trigger type of ba.*

| Methods | Cora $\mathcal{A}$ | $\mathcal{R}$ | $\mathcal{V}$ | PubMed $\mathcal{A}$ | $\mathcal{R}$ | $\mathcal{V}$ | CS $\mathcal{A}$ | $\mathcal{R}$ | $\mathcal{V}$ | Phy $\mathcal{A}$ | $\mathcal{R}$ | $\mathcal{V}$ | Amz $\mathcal{A}$ | $\mathcal{R}$ | $\mathcal{V}$ |
|---|---|---|---|---|---|---|---|---|---|---|---|---|---|---|---|
| Vanilla | 79.05 | 79.53 | 79.29 | 85.55 | 85.70 | 85.63 | 91.01 | 72.24 | 81.63 | 92.95 | 63.91 | 78.43 | 78.68 | 77.27 | 77.98 |
| FLTrust | 82.01 | 92.53 | 87.27 | 86.02 | 75.74 | 80.88 | 91.20 | 78.29 | 84.75 | 93.90 | 62.09 | 77.99 | 76.47 | 77.38 | 76.93 |
| RSA | 80.58 | 75.20 | 77.89 | 85.31 | 80.52 | 82.92 | 88.58 | 72.68 | 80.63 | 93.39 | 67.12 | 80.26 | 82.09 | 75.88 | 78.99 |
| RLR | 79.79 | 95.23 | **87.51** | 87.55 | 82.77 | 85.16 | 91.17 | 79.65 | 85.41 | 94.54 | 64.29 | 79.42 | 80.55 | 78.20 | 79.38 |
| FLAME | 77.79 | 89.83 | 83.81 | 86.78 | 74.52 | 80.65 | 88.58 | 78.41 | 83.50 | 94.04 | 58.45 | 76.24 | 76.43 | 79.10 | 77.77 |
| G²uard | 79.96 | 88.15 | 84.05 | 84.84 | 86.66 | 85.75 | 90.35 | 74.90 | 82.63 | 94.21 | 66.67 | 80.44 | 77.91 | 77.33 | 77.62 |
| Trim Median | 80.18 | 88.55 | 84.36 | 85.21 | 69.63 | 77.42 | 88.41 | 78.10 | 83.26 | 92.86 | 58.26 | 75.56 | 80.27 | 78.65 | 79.46 |
| Trimmed Mean | 78.67 | 82.70 | 80.68 | 86.35 | 75.63 | 80.99 | 90.21 | 76.89 | 83.55 | 93.58 | 62.03 | 77.81 | 80.99 | 79.10 | 80.05 |
| FreqFed | 81.44 | 92.75 | 87.10 | 85.87 | 80.21 | 83.04 | 90.69 | 76.59 | 83.64 | 94.44 | 60.57 | 77.50 | 81.44 | 78.23 | 79.84 |
| RFA | 81.92 | 89.38 | 85.65 | 85.70 | 77.47 | 81.59 | 89.33 | 75.60 | 82.47 | 93.68 | 64.73 | 79.20 | 81.32 | 79.01 | 80.17 |
| MMA | 80.97 | 92.53 | 86.75 | 85.67 | 78.27 | 81.96 | 87.97 | 77.81 | 82.89 | 92.34 | 60.50 | 76.42 | 78.67 | 76.88 | 77.78 |
| FoolsGold | 82.12 | 88.16 | 85.14 | 86.21 | 71.87 | 79.04 | 89.76 | 79.45 | 84.61 | 94.38 | 67.27 | 80.83 | 81.17 | 77.35 | 79.26 |
| DnC | 73.44 | 93.33 | 83.39 | 83.22 | 72.95 | 78.09 | 73.73 | 77.89 | 75.81 | 77.06 | 71.15 | 74.11 | 70.92 | 78.65 | 74.79 |
| FedCPA | 80.06 | 87.43 | 83.75 | 86.06 | 79.40 | 82.73 | 89.81 | 73.70 | 81.76 | 94.47 | 60.92 | 77.70 | 81.97 | 77.33 | 79.65 |
| Sageflow | 82.66 | 91.42 | 87.04 | 87.85 | 75.39 | 81.62 | 90.05 | 79.39 | 84.72 | 92.25 | 68.15 | 80.20 | 82.24 | 79.10 | 80.67 |
| FedTGE | 81.98 | 92.58 | 87.28 | 87.27 | 91.56 | **89.42** | 91.13 | 86.89 | **89.01** | 93.55 | 75.67 | **84.61** | 80.78 | 88.71 | **84.75** |

| Methods | Cora | | | PubMed | | | Coauthor-CS | | | Coauthor-Phy | | | Amz-Photo | | |
|---|---|---|---|---|---|---|---|---|---|---|---|---|---|---|---|
| | $\mathcal{A}$ | $\mathcal{R}$ | $\mathcal{V}$ | $\mathcal{A}$ | $\mathcal{R}$ | $\mathcal{V}$ | $\mathcal{A}$ | $\mathcal{R}$ | $\mathcal{V}$ | $\mathcal{A}$ | $\mathcal{R}$ | $\mathcal{V}$ | $\mathcal{A}$ | $\mathcal{R}$ | $\mathcal{V}$ |
| *Non-IID-Louvain with a malicious proportion of $\Upsilon = 0.3$ under a trigger type of ws.* | | | | | | | | | | | | | | | |
| Vanilla | 77.31 | 74.74 | 76.03 | 85.34 | 81.97 | 83.66 | 86.06 | 76.63 | 81.35 | 93.82 | 63.03 | 78.42 | 74.03 | 79.01 | 76.52 |
| FLTrust | 76.29 | 93.51 | 84.90 | 86.43 | 74.67 | 80.55 | 90.87 | 79.65 | 85.26 | 93.87 | 63.19 | 78.53 | 76.46 | 76.37 | 76.42 |
| RSA | 82.90 | 90.28 | 86.59 | 85.05 | 83.79 | 84.42 | 89.76 | 75.03 | 82.40 | 93.05 | 62.79 | 77.92 | 80.34 | 76.84 | 78.59 |
| RLR | 81.82 | 97.22 | 89.52 | 87.34 | 76.80 | 82.07 | 90.62 | 77.00 | 83.81 | 94.64 | 66.54 | 80.59 | 80.19 | 76.38 | 78.29 |
| FLAME | 82.25 | 84.23 | 82.24 | 86.26 | 70.37 | 78.32 | 89.32 | 79.45 | 84.39 | 93.37 | 58.83 | 76.10 | 77.51 | 79.10 | 78.31 |
| G$^2$uard | 77.99 | 94.86 | 86.42 | 85.32 | 78.74 | 82.03 | 90.04 | 79.29 | 84.67 | 93.43 | 69.18 | 81.31 | 77.22 | 75.01 | 76.12 |
| Trim Median | 77.31 | 74.74 | 76.03 | 85.13 | 71.39 | 78.26 | 90.22 | 77.79 | 84.01 | 92.95 | 58.41 | 75.68 | 80.82 | 77.75 | 79.29 |
| Trimmed Mean | 81.34 | 92.38 | **86.86** | 85.62 | 72.11 | 78.87 | 90.33 | 76.83 | 83.58 | 93.57 | 58.74 | 76.16 | 81.76 | 78.65 | 80.21 |
| FreqFed | 83.41 | 72.78 | 78.10 | 86.75 | 76.32 | 81.53 | 90.91 | 89.65 | 90.28 | 93.31 | 68.81 | 81.06 | 80.46 | 78.49 | 79.48 |
| RFA | 80.19 | 82.48 | 81.34 | 86.55 | 77.32 | 81.93 | 89.86 | 79.65 | 84.76 | 92.59 | 59.48 | 76.03 | 79.50 | 77.78 | 78.64 |
| MMA | 81.10 | 91.30 | 86.20 | 86.82 | 80.03 | 83.42 | 86.77 | 79.45 | 83.11 | 94.38 | 65.30 | 79.84 | 80.92 | 75.12 | 78.02 |
| FoolsGold | 82.91 | 84.72 | 83.82 | 87.02 | 78.28 | 82.65 | 90.91 | 78.12 | 84.52 | 94.09 | 68.29 | 81.19 | 80.07 | 88.68 | 84.38 |
| DnC | 64.89 | 86.40 | 75.65 | 78.75 | 72.18 | 75.46 | 77.75 | 96.44 | 87.10 | 78.34 | 78.43 | 78.38 | 66.78 | 98.78 | 82.78 |
| FedCPA | 82.20 | 90.50 | 86.35 | 85.81 | 79.09 | 82.45 | 90.24 | 79.65 | 84.95 | 94.57 | 69.83 | 82.20 | 81.02 | 76.84 | 78.93 |
| Sageflow | 81.81 | 89.40 | 85.61 | 86.10 | 86.34 | 86.22 | 89.54 | 76.78 | 83.16 | 94.00 | 67.24 | 80.62 | 77.76 | 73.26 | 75.51 |
| FedTGE | 81.76 | 91.98 | 86.87 | 86.23 | 90.19 | **88.21** | 90.83 | 92.96 | **91.90** | 94.45 | 85.69 | **90.07** | 78.69 | 97.12 | **87.91** |

## C DETAILS OF THE COMPARISON METHODS

- **FedAvg** (McMahan et al., 2017b): The standard federated averaging algorithm, where updates from all clients are averaged at the server without applying any defense mechanisms. This method is widely used in federated learning but is vulnerable to adversarial attacks.

- **Trimmed Median** and **Trimmed Mean** (Yin et al., 2018b): These methods are designed to mitigate the impact of malicious clients by trimming outliers or anomalous parameter updates. The trimmed median eliminates the extreme values from the client updates, while the trimmed mean excludes a proportion of the largest and smallest values before aggregation, thereby offering some protection against Byzantine failures.

- **FoolsGold** (Fung et al., 2018): A defense strategy aimed at preventing model poisoning attacks by reducing the weight of clients with highly similar updates. This method assumes that malicious clients tend to submit similar updates to amplify the impact of the backdoor attack, and thus it assigns lower aggregation weights to those clients.

- **DnC** (Shejwalkar & Houmansadr, 2021): This method partitions clients into different clusters based on their updates and then aggregates the updates within each cluster. By separating clients, DnC reduces the likelihood of adversarial clients overwhelming the aggregation process, making it harder for an attacker to influence the global model.

- **SageFlow** (Park et al., 2021): A distance-based defense method that detects and mitigates malicious updates by measuring the discrepancies between client updates. It ensures that updates that deviate significantly from the majority are either discarded or given lower aggregation weights, thus improving robustness against adversarial clients.

- **MMA** (Huang et al., 2023a): An adaptive defense mechanism that relies on multiple metrics, such as gradient norms, update similarities, and client performance, to identify and counteract adversarial behaviors. This method dynamically adjusts the aggregation weights of clients based on their performance across these metrics.

- **RFA** (Pillutla et al., 2022): This method uses the geometric median as a robust aggregation technique to handle outliers in client updates. The geometric median ensures that extreme values (e.g., from malicious clients) have less influence on the final aggregated model, thereby providing resilience against adversarial attacks.

- **RLR (Robust Learning Rate)** (Ozdayi et al., 2021): A defense approach that adaptively adjusts the learning rate of clients based on the perceived reliability of their updates. Clients that appear to submit suspicious or noisy updates receive lower learning rates, which helps in mitigating the influence of malicious clients and stabilizing the global model.

- **RSA** (Li et al., 2019a): RSA is a method designed to improve robustness in federated learning by introducing regularization into the aggregation process. It adjusts client updates by penalizing the updates that deviate significantly from the average, thereby mitigating the influence of malicious or noisy clients. The regularization term helps smooth out extreme variations in client contributions, ensuring a more stable and reliable global model. This method is particularly effective in environments with high heterogeneity, where individual client updates can vary greatly.

- **FreqFed** (Fereidooni et al., 2023): A frequency analysis-based defense mechanism designed to mitigate poisoning attacks in federated learning. FreqFed leverages frequency domain transformations (such as Fourier transforms) to analyze the client updates, identifying and filtering out malicious updates that exhibit anomalous frequency patterns. By decomposing the updates into their constituent frequencies, FreqFed can effectively distinguish between benign and adversarial modifications. This approach enhances the robustness of the global model by ensuring that only updates with consistent and expected frequency characteristics are aggregated, thereby reducing the impact of poisoning attacks and improving overall model integrity.

- **FedCPA** (Han et al., 2023): An attack-tolerant federated learning algorithm that performs critical parameter analysis to identify and mitigate the influence of malicious client updates. FedCPA operates by analyzing the importance of each parameter in the global model and selectively aggregating updates based on the criticality of the parameters. By focusing on the most critical parameters, FedCPA reduces the attack surface available to adversaries and enhances the robustness of the federated learning process. This method ensures that even if some clients are compromised,

their ability to significantly alter the global model is limited, thereby maintaining model integrity and performance.

- **FLTrust** (Cao et al., 2021): A Byzantine-robust federated learning framework that enhances robustness through trust bootstrapping. FLTrust assumes the server possesses a small, clean dataset referred to as the "root dataset" and maintains a trusted server model based on this dataset. In each training iteration, the server assigns trust scores to client updates based on their similarity to the server's update direction and normalizes the magnitude of client updates. The server then aggregates the normalized updates using a weighted averaging approach to update the global model. By leveraging a trusted root dataset, FLTrust effectively mitigates the impact of malicious client updates, improving the overall robustness of federated learning systems.

- **FLAME** (Nguyen et al., 2022): A defense framework designed to mitigate backdoor attacks in federated learning. FLAME estimates the required noise injection to eliminate backdoors while preserving the model's performance. To minimize the noise required, FLAME incorporates model clustering and weight pruning techniques. Experimental results demonstrate that FLAME effectively removes backdoor threats across various datasets, with negligible impact on the model's normal performance.

- **G$^2$uard** (Yu et al., 2023): A framework that safeguards federated learning against backdoor attacks through attributed client graph clustering. G$^2$uard reformulates the identification of malicious clients as an attributed graph clustering problem, leveraging client graph clustering methods to identify adversarial clients. It incorporates adaptive mechanisms to amplify differences between aggregated models and compromised models, effectively neutralizing embedded backdoors. Theoretical analyses confirm that G$^2$uard does not compromise the convergence of the federated learning system. Empirical results further validate its efficacy in significantly reducing attack success rates under various backdoor attack scenarios, while maintaining minimal impact on the model's performance on benign samples.

Table 4: **Statistics** of datasets used in experiments.

| Dataset | #Nodes | #Edges | #Classes | #Features |
|---------|--------|--------|----------|-----------|
| Cora | 2,708 | 5,278 | 7 | 1,433 |
| Pubmed | 19,717 | 44,324 | 3 | 500 |
| Coauthor-CS | 18,333 | 327,576 | 15 | 6,805 |
| Amz-Physics | 34,493 | 495,924 | 5 | 8415 |
| Amz-Photo | 7,650 | 287,326 | 8 | 745 |

## D  COMPLEXITY ANALYSIS

In this section, we delve into the computational complexity of FedTGE, which comprises two components, TEDC and TESP, analyzed separately. TEDC is employed to adjust the parameters of the BN layer, denoted as $P_{BN}$. Let $D$ represent the number of nodes, $F$ the number of features per node, and $E$ the number of edges. The energy calibration length is denoted as $EN$. The main computational complexity of TEDC can be divided into three parts: forward propagation for energy computation, energy gradient calculation, and directional propagation. The formulas are as follows:

$$O(E \times F) + O(D \times F) + O(D) + O(P_{BN}) \tag{16}$$

In non-dense graphs, $E$ can be considered proportional to $D$. Considering energy calibration, the formula can be simplified as:

$$O(E \times F \times EN) \tag{17}$$

We define the number of clients as $N$, the indexes established between clients as $E$, and the energy distribution length as $L$. The computational complexity of TESP mainly arises from similarity matrix computation, edge indexing, and energy propagation, which can be formalized as:

$$O(N^2 \times L) + O(N^2) + O(N) + O(E) + O(K \times E) \tag{18}$$

Since $N$ and $E$ are usually relatively small constants, the above formula can be further simplified as:

$$O(D + K) \tag{19}$$

This indicates that the two components of our proposed FedTGE scale linearly with the number of nodes or edges, demonstrating its suitability for large-scale datasets.

## E  VISUALIZATION RESULTS

To provide a clearer view of TEDC's energy correction effect, we visualize the energy distribution of malicious and benign clients across multiple datasets under the non-iid-louvain setting from the perspective of **Kernel Density Estimation Plot** (Davis et al., 2011; Parzen, 1962). The default settings for all visualization experiments are as follows: the total number of clients is 10, with 3 designated as malicious; the trigger size is set to 4; the poisoning intensity is 0.3; the energy epoch is 10; and the trigger types include renyi (Zhang et al., 2021c), GTA (Xi et al., 2021), BA (Barabási & Albert, 1999), and WS (Watts & Strogatz, 1998).

## E.1 CORA

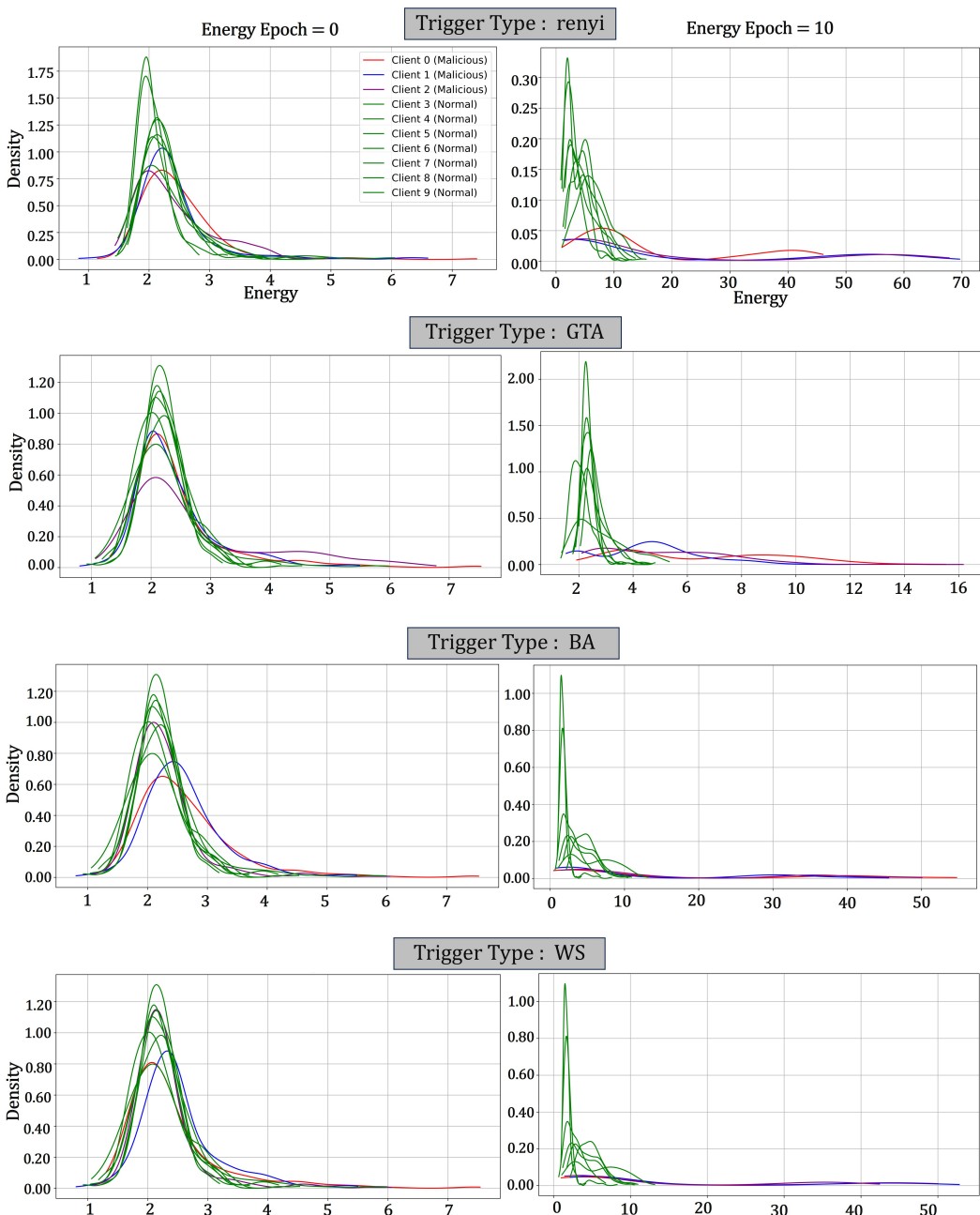

## E.2 PUBMED

Trigger Type : renyi

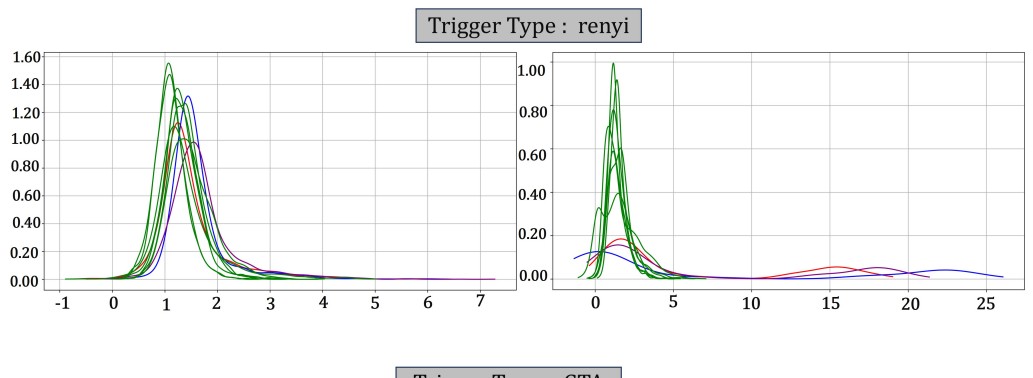

Trigger Type : GTA

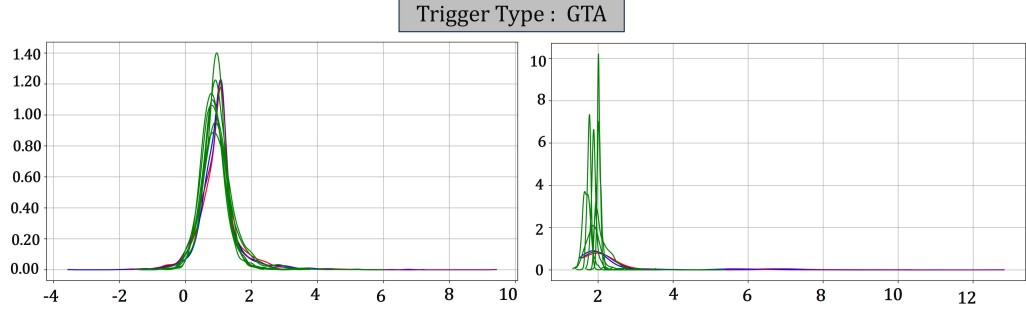

Trigger Type : BA

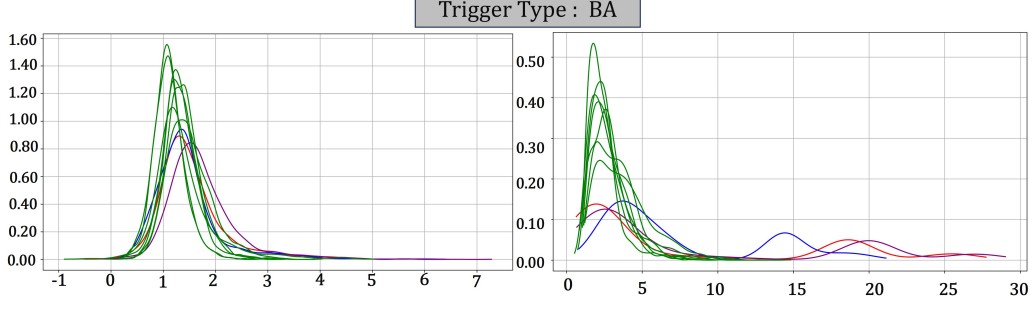

Trigger Type : WS

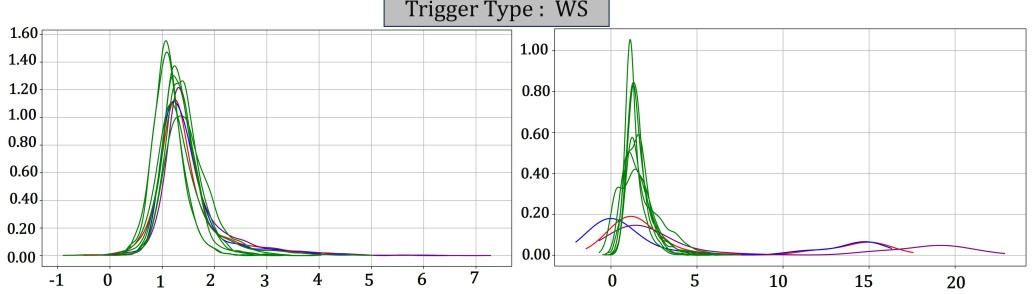

## E.3 PHOTO

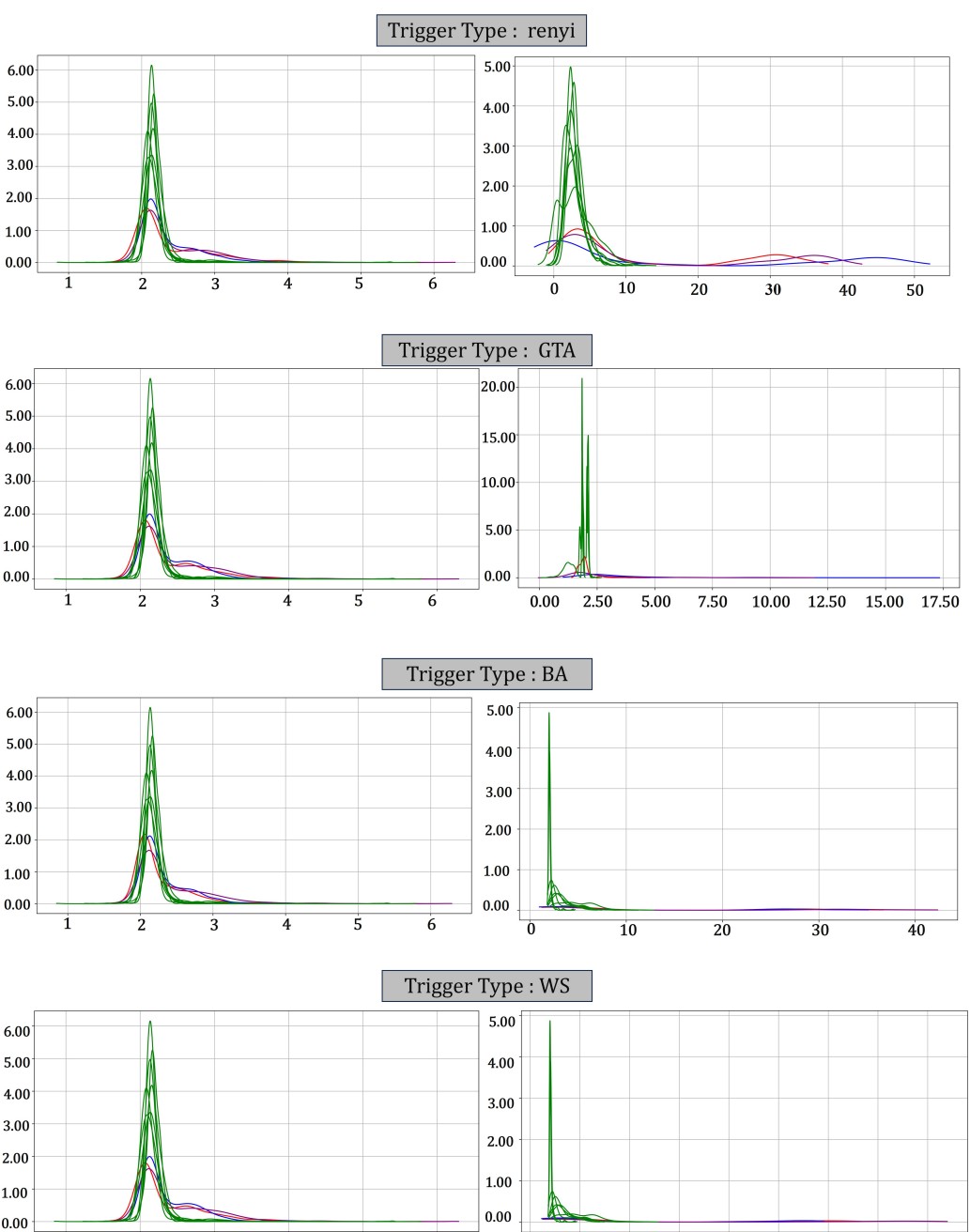

## E.4    Cs

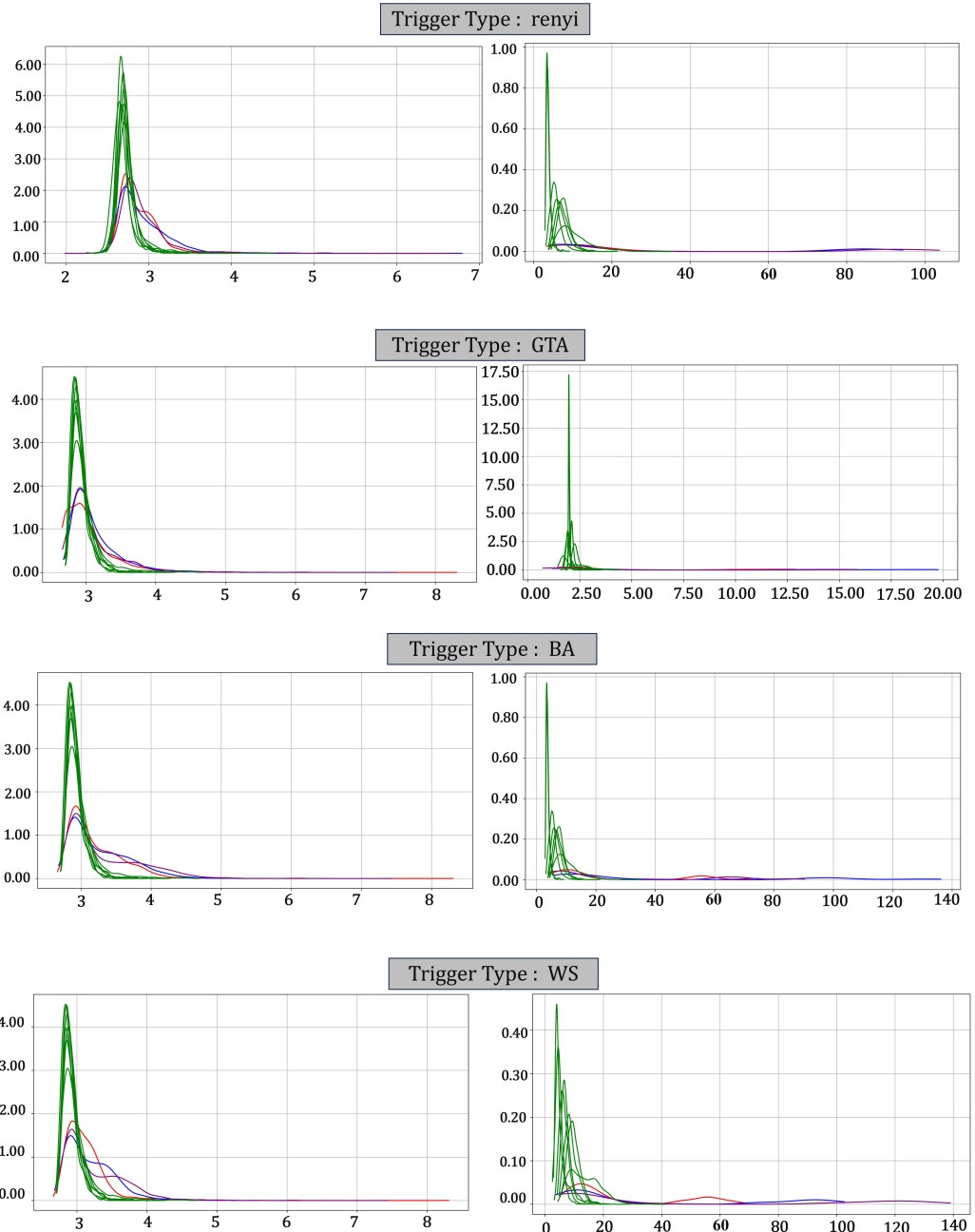

## E.5 PHYSICS

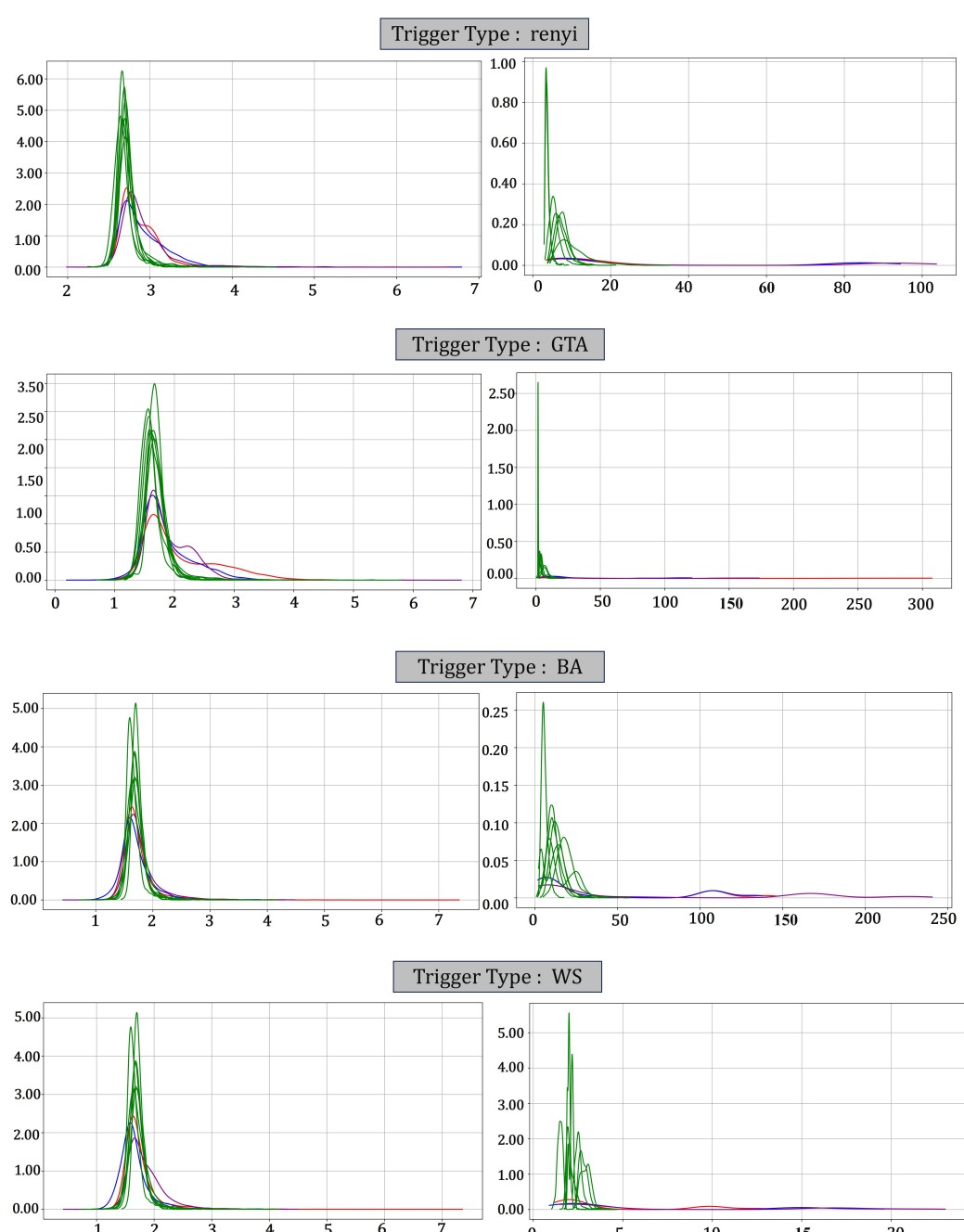

## F VISUALIZATION OF FEDFREQ

In the non-iid-louvain environment, we used Principal Component Analysis (PCA) to visualize the low-frequency components extracted by FedFreq. The clustering results were marked with red circles to highlight the identified groups.

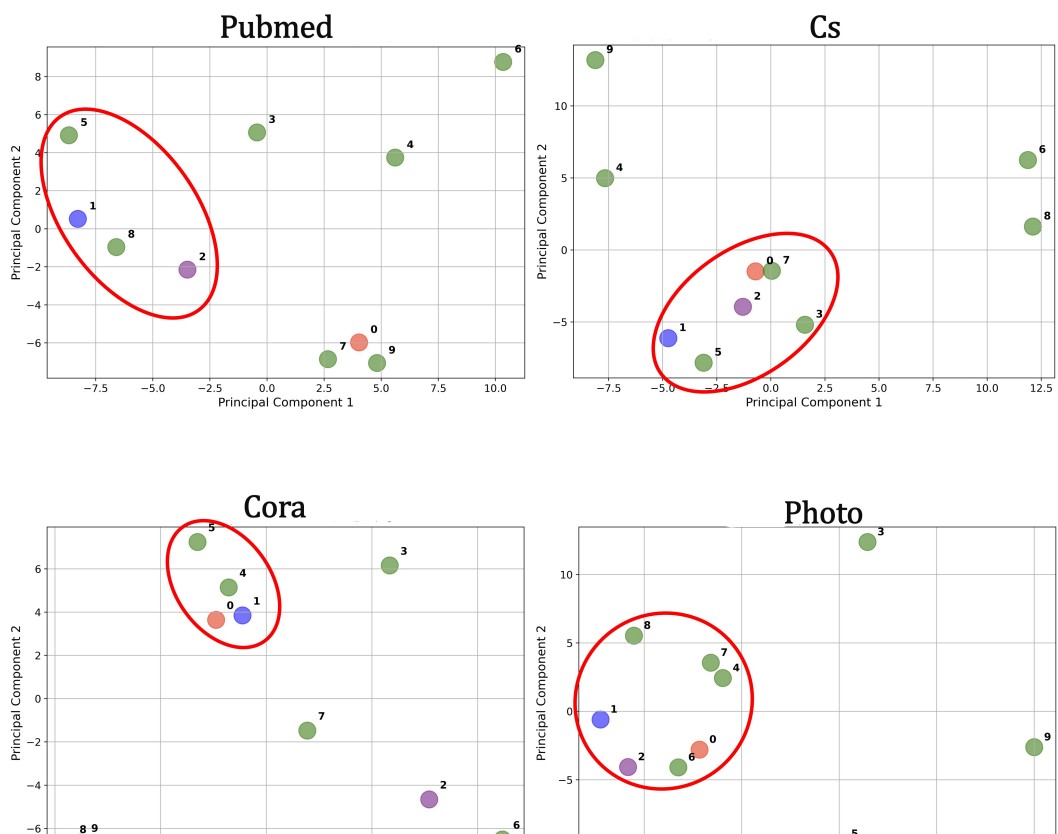

Visualization reveals that the low-frequency component-based defense proposed by FedFreq is not suitable for GCNs, as malicious and benign updates are intertwined, and even benign clients are separated from each other. This limitation prevents FedFreq from correctly filtering malicious updates and may even discard updates from some benign clients, potentially leading to performance degradation.

