# OpenReview forum: "Energy-based Backdoor Defense Against Federated Graph Learning"
_ICLR.cc/2025/Conference — ICLR 2025 Oral_

### Official Review · Reviewer_eV6Q · 2024-10-23

**Soundness:** 3
**Presentation:** 3
**Contribution:** 3
**Rating:** 8
**Confidence:** 5

**Summary:**

The authors propose an energy-based backdoor defense for Federated graph learning. Experimental results on the node classification task under the mentioned backdoor attacks are provided, with comparisons to several defenses, and performance gain are observed in iid and non-iid scenarios.

**Strengths:**

1. The estimation of using energy distribution to recognize malicious clients seems to be novel.

2. Comparisons with some advanced defenses are provided and show the advantages.

**Weaknesses:**

1. Key details on the different energy distributions of malicious clients and others are not provided. In addition, the influence of this factor in the non-iid scenario needs further verification.

2. Some standard defenses are missing, such as FLAME， FL Trust, and G^2uard Fl. In addition, the authors should pay attention to their statements carefully, since the proposed work actually is not the first to conduct backdoor attacks in FGL. For example, "Distributed backdoor attacks on federated graph learning and certified defenses" published in CCS 2024 is a good work that should be compared.

3. It is strange that FedFreq has an unexpectedly low performance in the experiments which are not consistent in their own work, and the authors may clarify the reason.

**Questions:**

Besides the weakness, the authors may consider the following points to enhance the paper.

1. Does the proposed energy-based estimation work on traditional FL or other domains?

2. Does the proposed scheme work for untargeted attacks?

3. Does the proposed scheme work for other types of graph learning tasks, such as graph classification and link prediction?

4. The verification attacking method needs more details, and other attacks should be further evaluated.

---

> ### Author Response · Authors · 2024-11-21
> **[Part 1/3] Response to the Reviewer eV6Q**
>
> Dear Reviewer ev6Q
>
> We sincerely thank you for your insightful feedback. We have provided detailed responses to your questions, and we hope that these clarifications and revisions meet your expectations and potentially merit a higher evaluation.
>
> > `Weakness 1`: Visualization and More non-iid scenarios
>
> Thank you for your valuable feedback regarding the details of the experiments. To address this issue comprehensively, we have provided detailed visualizations of the energy distribution for each client before and after clustering, as well as additional experimental reports for non-iid scenarios. These have included in Appendix E and Appendix B, respectively. Below we present additional experimental results for non-IID scenarios:
>
> **renyi** **(Non-iid-feature-skew with alpha = 0.5 and a malicious proportion of 0.3)** :
>
> |   Methods   |     Cora  (A, R, V)     |    Pubmed  (A, R, V)    |   Physics  (A, R, V)    |
> | :---------: | :-------------------------: | :-------------------------: | :-------------------------: |
> |   FedAvg    |   62.97,   44.79,   53.88   |   84.00,   31.60,   57.80   |   94.05,   48.23,   71.14   |
> |   FLTrust   |   60.47,   55.56,   58.02   |   83.68,   32.17,   57.93   |   94.55,   55.36,   74.96   |
> |  FoolsGold  |   66.38,   43.21,   54.79   |   86.10,   35.35,   60.73   |   94.02,   44.58,   69.30   |
> |    FLAME    |   60.30,   66.50,   63.40   |   83.48,   62.34,   72.91   |   92.85,   62.32,   77.59   |
> |  Sageflow   |   65.98,   56.37,   61.18   |   87.99,   60.24,   74.12   |   93.81,   69.98,   81.90   |
> | **FedTGE**  | 63.69,   88.98,   **76.34** | 86.53,   71.29,   **78.91** | 94.40,   85.70,   **90.05** |
>
> **renyi** **(Non-iid-label-skew with alpha = 0.5 and a malicious proportion of 0.3)** :
>
> |   Methods   |     Cora  (A, R, V)     |    Pubmed  (A, R, V)    |   Physics  (A, R, V)    |
> | :---------: | :-------------------------: | :-------------------------: | :-------------------------: |
> |   FedAvg    |   61.72,   73.89,   67.80   |   85.88,   11.28,   48.58   |   94.34,   39.54,   69.93   |
> |   FLTrust   |   63.12,   59.66,   61.39   |  82.63,    8.98,    45.80   |   94.58,   31.25,   62.92   |
> |  FoolsGold  |   65.22,   63.98,   64.60   |   86.86,   10.91,   48.88   |   91.31,   61.05,   77.18   |
> |    FLAME    |   63.27,   70.08,   66.68   |   84.28,   14.43,   49.36   |   90.27,   42.64,   66.46   |
> |  Sageflow   |   65.42,   44.91,   55.17   |  86.09,    0.35,    43.22   |   94.59,   26.51,   60.55   |
> | **FedTGE**  | 64.69,   88.91,   **76.60** | 86.98,   43.65,   **65.32** | 94.79,   86.79,   **90.79** |
>
> ---
>
> > `Weakness 2`: Omission of Standard Defenses
>
> We appreciate your valuable feedback and would like to point out the omission of certain standard defense methods in our initial submission. We have carefully reviewed [1] and have incorporated the methods you mentioned as references in the revised manuscript. What we want to clarify is that FLTrust relies on the availability of a clean and trustworthy dataset at the server side. However, the size or feasibility of **such a dataset has not been clearly established in graph domain**. And this limitation was the reason it was not it was not included in the original experimental design.
>
> [1] is an excellent concurrent work focusing on backdoor attacks and defenses in the context of graph classification. The Certified Defense method discussed in [1] primarily involves dividing test graphs into multiple subgraphs and determining the final label through majority voting. However, this approach does not fully adapt to node classification tasks. In contrast, our proposed FedTGE performing node-level energy adjustment is also applicable to defense tasks in graph classification scenarios. For relevant experimental results, please refer to our response to Question 3.
>
> ---
>
> > `Weakness 3`: Reasons for the performance degradation of FedFreq
>
> Thank you for highlighting this concern. FedFreq is indeed an advanced method in traditional federated learning, relying on clustering and filtering malicious clients based on their uploaded model updates. However, backdoor attack patterns in FGL pose greater challenges compared to those in traditional FL. Specifically, the randomness in the location and shape of trigger injections, as well as their distribution, makes traditional model update-based defense methods, such as Trimmed Mean and FoolsGold, less effective in FGL.
>
> For FedFreq, it relies on model updates to identify malicious clients and completely excludes the model parameters of these clients. However, this approach often fails to accurately identify malicious clusters in FGL. This inevitably leads to the loss of knowledge learned by some clients, resulting in significant performance degradation in certain scenarios. A similar issue can also be observed in the DnC algorithm.
>
> ---

---

> > ### Author Response · Authors · 2024-11-21
> > **[Part 2/3] Response to the Reviewer eV6Q**
> >
> > > `Question 1`: Does the proposed energy-based estimation work on traditional FL or other domains?
> >
> > Thank you for this question. The core of the FedTGE method lies in modeling and adjusting the energy of samples (i.e., their unnormalized probabilities). Based on this insight, energy-based estimation can be extended to a variety of fields. This includes traditional FL, such as backdoor defense, where it can similarly assign lower energy to benign samples. However, since images lack the structural properties inherent to graphs, its effectiveness in this domain may be limited. Additionally, it can also be utilized to enhance model generalization. For instance, [2] is an excellent related work that improves model generalization by adjusting the energy of test samples.
> >
> > ---
> >
> > > `Question 2`: Does the proposed scheme work for untargeted attacks?
> >
> > An untargeted attack does not focus on specific target nodes, instead, its impact typically spans the entire graph or a large subset of samples, often significantly disturbing the statistical distribution of the input data. Although our method is specifically designed for targeted attacks, it can theoretically also be applied to untargeted attacks. To demonstrate this, we conducted a simple experiment under the DICE [3] with 10% perturbed edges setting, using the misclassification rate (%) as the evaluation metric. The results are as follows:
> >
> > |      Methods      |    Pubmed    |    photo     |      Cs      |
> > | :---------------: | :----------: | :----------: | :----------: |
> > |     **Clean**     | 15.26 ± 0.35 | 25.14 ± 0.78 | 9.78 ± 0.45  |
> > |     **DICE**      | 19.11 ± 1.12 | 30.28 ± 1.33 | 15.78 ± 1.12 |
> > | **DICE + FedTGE** | 16.21 ± 0.37 | 26.23 ± 0.52 | 9.83 ± 0.34  |
> >
> > ---
> >
> >
> >
> > > `Question 3`: Does the proposed scheme work for other types of graph learning tasks, such as graph classification and link prediction?
> >
> > Thank you for this question. We have also applied the proposed method to graph classification and link prediction to further verify its scalability. The results are shown below, where the metrics are reported following the original manuscript's A, R, and V:
> >
> > **graph classification**
> >
> > |            Methods            |     NCI1 (A,R,V)      | PROTEINS_full  (A,R,V)  |       DD  (A,R,V)       |
> > | :---------------------------: | :-------------------------: | :-------------------------: | :-------------------------: |
> > |          FedAvg           |   81.22,   17.59,   49.41   |   73.84,   48.61,   61.23   |   63.84,   16.88,   40.36   |
> > |       Trimmed Mean        |   80.78,   51.59,   66.19   |   74.21,   48.60,   61.41   |   62.04,   11.43,   36.74   |
> > |        Trim Median        |   79.51,   30.70,   55.11   |   74.77,   48.61,   61.69   |   64.11,   18.33,   41.22   |
> > |            RSA            |   78.70,   20.15,   45.43   |   73.34,   47.89,   60.62   |   66.35,   11.13,   38.74   |
> > |           FLAME           |   77.62,   37.19,   57.41   |   73.60,   50.12,   61.86   |   66.18,   32.25,   49.22   |
> > |          G^2uard          |   78.70,   42.26,   60.48   |   74.40,   49.21,   61.81   |   64.51,   13.03,   38.77   |
> > |         Sageflow          |   76.67,   27.48,   52.08   |   74.79,   47.38,   61.09   |   66.11,   14.07,   40.09   |
> > |          FreqFed          |   76.60,   40.93,   58.77   |   76.05,   52.78,   64.42   |   61.24,   14.46,   37.85   |
> > | Certified Defense (css24) |   79.64,   64.19,   71.92   |   74.04,   53.16,   63.60   |   64.39,   42.77,   53.58   |
> > |          FedTGE           | 80.13,   64.89,   **72.51** | 75.29,   61.23,   **68.26** | 65.89,   47.62,   **56.76** |
> >
> >
> > **link prediction**
> >
> > The attack methods for link prediction are highly similar to those for node classification, as both involve injecting fake nodes or subgraphs to alter the contextual information of local graph data. Therefore, FedTGE is theoretically fully applicable to link prediction tasks.

---

> > > ### Author Response · Authors · 2024-11-21
> > > **[Part 3/3] Response to the Reviewer eV6Q**
> > >
> > > > `Question 4`: The verification attacking method needs more details, and other attacks should be further evaluated.
> > >
> > > Thank you for pointing this out. In the revised manuscript, we have included more details about the attacking method and experimental results for different trigger patterns in Appendix B.We have selected results for some diverse trigger structures and presented below.
> > >
> > > **GTA (Non-iid-louvain with a malicious proportion of 0.3):**
> > >
> > > |   Methods   |     Cora  (A, R, V)     |    Pubmed  (A, R, V)    |   Physics  (A, R, V)    |
> > > | :---------: | :-------------------------: | :-------------------------: | :-------------------------: |
> > > |   FedAvg    |   78.54,   27.96,   53.25   |   86.86,   11.70,   49.28   |   93.42,   24.86,   59.14   |
> > > |   FLTrust   |   78.21,   43.01,   60.61   |   86.36,   36.05,   61.21   |   94.21,   59.26,   76.74   |
> > > |  FoolsGold  |   79.56,   25.60,   52.58   |   89.15,   16.70,   52.87   |   94.68,   40.56,   67.62   |
> > > |    FLAME    |   77.54,   67.13,   72.34   |   85.45,   40.51,   62.98   |   93.28,   50.49,   71.89   |
> > > | Trim Median |   79.49,   33.06,   56.26   |   86.63,   14.58,   50.60   |   93.55,   27.39,   60.47   |
> > > |  Sageflow   |   79.95,   29.49,   54.72   |   89.04,   16.70,   52.87   |   94.54,   54.69,   74.62   |
> > > | **FedTGE**  | 80.23,   75.23,   **77.73** | 87.56,   55.34,   **71.45** | 94.39,   72.37,   **83.38** |
> > >
> > > **WS (Non-iid-louvain with a malicious proportion of 0.3):**
> > >
> > > |   Methods   |     Cora  (A, R, V)     |    Pubmed  (A, R, V)    |   Physics  (A, R, V)    |
> > > | :---------: | :-------------------------: | :-------------------------: | :-------------------------: |
> > > |   FedAvg    |   77.31,   74.74,   76.03   |   85.34,   81.97,   83.66   |   93.82,   63.03,   78.42   |
> > > |   FLTrust   |   76.29,   93.51,   84.90   |   86.43,   74.67,   80.55   |   93.87,   63.19,   78.53   |
> > > |  FoolsGold  |   82.91,   84.72,   83.82   |   87.02,   78.28,   82.65   |   94.09,   68.29,   81.19   |
> > > |    FLAME    |   82.25,   84.23,   82.24   |   86.26,   70.37,   78.32   |   93.37,   58.83,   76.10   |
> > > | Trim Median |   77.31,   74.74,   76.03   |   85.13,   71.39,   78.26   |   92.95,   58.41,   75.68   |
> > > |  Sageflow   |   81.81,   89.40,   85.61   |   86.10,   86.34,   86.22   |   94.00,   67.24,   80.62   |
> > > | **FedTGE**  | 81.76,   91.98,   **86.87** | 86.23,   90.19,   **88.21** | 94.45,   85.69,   **90.07** |
> > >
> > > The GTA algorithm injects triggers with distinct shapes, such as star or ring patterns, which exhibit high attack efficacy but suffer from low stealthiness. In contrast, the WS algorithm based on the Watts-Strogatz small-world model introduces triggers that typically possess realistic network characteristics resulting in higher stealthiness but at the cost of relatively lower attack effectiveness. Nevertheless, FedTGE is able to maintain strong defense performance even under these attack scenarios.
> > >
> > >
> > >
> > > [1] : Yang, Y.; Li, Q.; Jia, J.; Hong, Y.; and Wang, B. 2024. Distributed Backdoor Attacks on Federated Graph Learning and Certified Defenses.
> > >
> > > [2] : Xiao, Z.; Zhen, X.; Liao, S.; and Snoek, C. G. M. 2024. Energy-Based Test Sample Adaptation for Domain Generalization.
> > >
> > > [3] : Hristozov, S.; Wettermann, M.; and Huber, M. 2022. A TOCTOU Attack on DICE Attestation.

---

> > > ### Comment · Reviewer_eV6Q · 2024-11-21
> > >
> > > For Question 3, (linked to the authors' response for weakness 2), it shows the proposed algorithm has a better performance than Certified Defense (ccs 24), which is unexpected. To extend the proposed method into the graph classification scenario, what kind of modifications should be made? In addition, the implementation details should be provided for further verification.

---

> > ### Comment · Reviewer_eV6Q · 2024-11-21
> >
> > For weakness 1, it would be better to show the connection between the different energy distributions with node distribution. In addition, it seems that there is no appendix D in the submitted file.
> >
> > For weakness 2, the authors are suggested to refine the statement since the authors have not clearly clarified the paper only focuses on the node classification problem in Section 1.
> >
> > For weakness 3, thanks for the response. However, this explanation is not actually clear why FedFreq fails, and why it cannot identify malicious clusters.

---

> ### Author Response · Authors · 2024-11-21
> **Thank you for your response to our rebuttal**
>
> > `Weakness 1 & 2`
>
> Thank you for pointing out issues that were not clearly explained. We apologize for any inconvenience caused by the delay in updating it and the submission revision has been uploaded now.  The visualization of the energy distribution before and after energy calibration is provided in Appendix E. We have explicitly stated in the contributions section that our work focuses on node classification and is extendable to graph classification.
>
>
>
> > `Weakness 3`: Reasons for the performance degradation of FedFreq
>
> FedFreq assumes that the low-frequency components of model parameter weights represent the overall trend and global characteristics of weight changes, while high-frequency components are associated with noise and local details. However, the weight distribution of graph data is highly dependent on the graph's topological structure (e.g., node degree distribution and community structure). In non-Euclidean spaces, the graph topologies of different clients can vary significantly, leading to completely different spectral distributions even among benign clients. This variability makes it difficult to accurately distinguish between malicious and benign updates based on low-frequency features. As a result, HDBSCAN may misclassify the weights of benign clients as malicious updates.
>
>
>
> In the revised manuscript, we have added a visualization analysis of the low-frequency components of FedFreq in Appendix F. We use Principal Component Analysis (PCA) to visualize the distances of low-frequency components across different clients. It can be observed that defense methods based on low-frequency components (model updates) are not suitable for GCNs and may even filter out benign clients.

---

> ### Author Response · Authors · 2024-11-21
> **Thank you for your response to our rebuttal**
>
> > `Question 3`
>
> The modifications required to apply FedTGE to graph classification focus on replacing the perturbation of node subgraphs with the perturbation of entire graphs. Additionally, the defense mechanism of Certified Defense (CCS 24) may have inherent limitations, which could account for its inferior performance compared to FedTGE: (1) **Loss of Graph Information**: Certified Defense splits the test graph into multiple non-overlapping subgraphs for individual predictions. This process can result in significant information loss, potentially affecting prediction accuracy. For example, in domains like proteins or biomolecules, arbitrary partitioning of graphs may fail to produce correct labels due to the critical interdependencies within the graph structure. (2) **Lower Robustness**: Certified Defense cannot directly guarantee that the trigger will be entirely contained within a single subgraph during the partitioning process. If the trigger is distributed across multiple subgraphs, it could influence the predictions of several subgraphs, making the voting mechanism appear fragile and less reliable. These issues ultimately result in Certified Defense achieving overall performance that is inferior to FedTGE.

---

> > ### Comment · Reviewer_eV6Q · 2024-11-21
> >
> > Thanks for the detailed response. The reviewer has no further comments and will increase the score.

---

> > > ### Author Response · Authors · 2024-11-21
> > >
> > > Thank you for your thoughtful consideration and willingness to reevaluate the score. We sincerely appreciate your recognition of our efforts and the constructive feedback that has helped us improve our work.

---

### Official Review · Reviewer_i7KZ · 2024-11-01

**Soundness:** 3
**Presentation:** 3
**Contribution:** 4
**Rating:** 8
**Confidence:** 4

**Summary:**

This paper addresses the issue of backdoor attacks in Federated Graph Learning and proposes a defense method based on Topological Graph Energy. The method injects structural distributional knowledge into the model. At the client level, it models energy to differentiate between benign and malicious samples. While at the server level, it constructs a global energy graph for energy propagation, effectively identifying and excluding malicious clients. Experimental results demonstrate that FedTGE performs well under various attack proportions and in heterogeneous scenarios.

**Strengths:**

- The paper is the first to introduce energy-based models into backdoor defense in FGL, effectively addressing the complexities of trigger injection locations and diverse attack forms by learning the energy structure distribution of the data through constructing an energy-based model, which leverages the unique topological characteristics of graph data.
- The methodology is well explained and structured. In particular, the authors provide a detailed step-by-step breakdown of their approach to help in understanding the core mechanisms of FedTGE. Furthermore, the framework diagram clearly explains the energy propagation and clustering processes, making it easier to grasp the interactions between different modules.
- By constructing an energy graph and propagating energy across clients, FedTGE overcomes the limitations of traditional federated defense methods struggle to adapt well of graph data. This approach combines structural knowledge to enable more fine-grained and accurate identification of malicious entities. Thus enhancing the robustness and adaptability of the defense system across diverse scenarios.

**Weaknesses:**

- Although the experiments demonstrate the superiority of FedTGE over traditional methods, the paper lacks sufficient discussion of existing backdoor defense techniques in traditional federated learning. For instance, RFA appears in the performance table but is not comparatively analyzed in the introduction. The authors are supposed to clarify the shortcomings of these methods in the given scenario.

**Questions:**

I noticed that the authors conducted extensive experiments under the "Renyi" setting and achieved excellent results. However, to my knowledge, there are other effective attack methods in FGL, such as GTA. Could the authors provide relevant experiments to verify the effectiveness of their approach against such attacks?

**Details Of Ethics Concerns:**

NO.

---

> ### Author Response · Authors · 2024-11-21
> **[Part 1/1] Response to the Reviewer i7KZ**
>
> Dear Reviewer i7k7
>
> We sincerely thank you for taking the time to evaluate our work and have adressed your concerns as follows:
>
> > `Weakness 1`: More discussion on the baseline.
>
> We appreciate your feedback recognizing the advantages of our approach and pointing out the missing discussion on certain baselines. We would like to clarify that, in Section 5.2 (Performance Comparison), we categorized RFA under statistical distribution-based methods and provided an explanation. To ensure a comprehensive analysis of why baselines may not perform well in the given scenario, we will provide further details or clarifications as needed to enhance your understanding of our work.
>
> > `Question 1`: More defense performance under various trigger types
>
> We sincerely appreciate your insightful observations. In the revised manuscript, we have provided additional experimental reports in Appendix B. We have selected some of the results to display below:
>
> **GTA (Non-iid-louvain with a malicious proportion of 0.3):**
>
> |   Methods   |     Cora  (A, R, V)     |    Pubmed  (A, R, V)    |   Physics  (A, R, V)    |
> | :---------: | :-------------------------: | :-------------------------: | :-------------------------: |
> |   FedAvg    |   78.54,   27.96,   53.25   |   86.86,   11.70,   49.28   |   93.42,   24.86,   59.14   |
> |   FLTrust   |   78.21,   43.01,   60.61   |   86.36,   36.05,   61.21   |   94.21,   59.26,   76.74   |
> |  FoolsGold  |   79.56,   25.60,   52.58   |   89.15,   16.70,   52.87   |   94.68,   40.56,   67.62   |
> |    FLAME    |   77.54,   67.13,   72.34   |   85.45,   40.51,   62.98   |   93.28,   50.49,   71.89   |
> | Trim Median |   79.49,   33.06,   56.26   |   86.63,   14.58,   50.60   |   93.55,   27.39,   60.47   |
> |  Sageflow   |   79.95,   29.49,   54.72   |   89.04,   16.70,   52.87   |   94.54,   54.69,   74.62   |
> | **FedTGE**  | 80.23,   75.23,   **77.73** | 87.56,   55.34,   **71.45** | 94.39,   72.37,   **83.38** |
>
> **WS (Non-iid-louvain with a malicious proportion of 0.3):**
>
> |   Methods   |     Cora  (A, R, V)     |    Pubmed  (A, R, V)    |   Physics  (A, R, V)    |
> | :---------: | :-------------------------: | :-------------------------: | :-------------------------: |
> |   FedAvg    |   77.31,   74.74,   76.03   |   85.34,   81.97,   83.66   |   93.82,   63.03,   78.42   |
> |   FLTrust   |   76.29,   93.51,   84.90   |   86.43,   74.67,   80.55   |   93.87,   63.19,   78.53   |
> |  FoolsGold  |   82.91,   84.72,   83.82   |   87.02,   78.28,   82.65   |   94.09,   68.29,   81.19   |
> |    FLAME    |   82.25,   84.23,   82.24   |   86.26,   70.37,   78.32   |   93.37,   58.83,   76.10   |
> | Trim Median |   77.31,   74.74,   76.03   |   85.13,   71.39,   78.26   |   92.95,   58.41,   75.68   |
> |  Sageflow   |   81.81,   89.40,   85.61   |   86.10,   86.34,   86.22   |   94.00,   67.24,   80.62   |
> | **FedTGE**  | 81.76,   91.98,   **86.87** | 86.23,   90.19,   **88.21** | 94.45,   85.69,   **90.07** |
>
> The GTA algorithm injects triggers with distinct shapes, such as star or ring patterns, which exhibit high attack efficacy but suffer from low stealthiness. In contrast, the WS algorithm based on the Watts-Strogatz small-world model introduces triggers that typically possess realistic network characteristics resulting in higher stealthiness but at the cost of relatively lower attack effectiveness. Nevertheless, FedTGE is able to maintain strong defense performance even under these attack scenarios.
>
>
>
> ---

---

> > ### Comment · Reviewer_i7KZ · 2024-11-21
> >
> > I have reviewed the rebuttal and am satisfied that the authors have adequately addressed my main concerns. Additionally, I have reviewed the opinions of the other reviewers, and while most raised minor concerns about the experiments, there is general agreement that the proposed method is both innovative and well-executed, and the paper is clearly presented. Considering these strengths, I believe this paper stands out as an excellent submission for ICLR. I am inclined to maintain my score and clearly support its acceptance.

---

> > > ### Author Response · Authors · 2024-11-24
> > >
> > > ### Dear Reviewer i7KZ,
> > > We greatly appreciate your thoughtful feedback and unwavering support for our work. Your detailed suggestions have played a crucial role in refining our study. We deeply value the effort and expertise you have dedicated to this review process, and we remain truly grateful for your guidance and support.
> > >
> > > Best regards,
> > >
> > > Authors

---

### Official Review · Reviewer_8A86 · 2024-11-04

**Soundness:** 3
**Presentation:** 3
**Contribution:** 3
**Rating:** 8
**Confidence:** 5

**Summary:**

This paper presents FedTGE, a new method to protect Federated Graph Learning from backdoor attacks by analyzing energy patterns in data distribution. The method works by identifying and separating benign and malicious clients and adjusts how their data contribute to the final FL model. The authors claim that FedTGE performs better than current methods, which can handle high data heterogeneity, does not require a validation dataset.

**Strengths:**

-This topic is very important to the community, considering the backdoor defense methodology is developing.

-Well written and interesting.

-Thorough empirical results over a plethora of FL methods.

**Weaknesses:**

I have some concerns about the assumption and evaluation of this paper below.

**Questions:**

1. The paper assumes that calculating the energy distribution of clients can effectively tell malicious and benign clients apart. However, this method relies on the energy model accurately capturing the real data distribution. If the energy model fails to do so, especially in noisy or structurally complex data, the assumption is doubtful.

2. While the paper claims the method is effective under non-IID scenarios, I am not confident if the evaluation under non-IID-louvain setting is enough. For example,  there can be node label distribution skew and node feature skew. More non-IID conditions should be evaluated.

3.  TESP is used to adjust aggregation weights to defend against malicious clients. How to prevent from incorrectly including benign clients?

4. The computational complexity of the energy model and similarity propagation might become a bottleneck in large-scale cases, the authors should discuss this issue.

---

> ### Author Response · Authors · 2024-11-21
> **[Part 1/2] Response to the Reviewer 8A86**
>
> Dear Reviewer 8A86
>
> Thank you for recognizing the contribution of our work to the community. Your valuable input has been instrumental in helping us refine and enhance our research. Please find our detailed responses below:
>
> > `Question 1`: The reasonableness of the assumption
>
> We understand your concern regarding this issue. Our energy model is essentially an energy-based model that integrates the ability of GNNs to capture data structures while attaching energy information to the graph data. Samples that align well with the data modeled by GNNs are assigned low energy, whereas those that do not are assigned high energy.
>
> In datasets with low noise and simple structures, these triggers significantly alter the structural information of the data, serving as a prominent signal for FedTGE to assign higher energy. For example, in the IID scenarios, FedTGE achieves excellent defense performance. Furthermore, in datasets with moderate noise, FedTGE remains effective : **(1)**  TESP facilitates repeated energy propagation, narrowing the energy distribution gap among selected clients, which further enhances the clustering effect of TEDC. The threshold filtering mechanism also ensures that mistakenly selected malicious clients are assigned lower aggregation weights or even excluded entirely. **(2)** For triggers, to achieve effective attacks in noisy environments, they must undergo corresponding adjustments, such as adopting more complex trigger structures or incorporating features that deviate from the context. These characteristics assist TEDC and TESP in filtering them out effectively. Our extensive experiments also validate that under the same noise levels, FedTGE consistently outperforms the baseline methods. Below we present a portion of the results from the feature skew experiments.
>
> **renyi** **(Non-iid-feature-skew with alpha = 0.5 and a malicious proportion of 0.3)** :
>
> |   Methods   |     Cora  (A, R, V)     |    Pubmed  (A, R, V)    |   Physics  (A, R, V)    |
> | :---------: | :-------------------------: | :-------------------------: | :-------------------------: |
> |   FedAvg    |   62.97,   44.79,   53.88   |   84.00,   31.60,   57.80   |   94.05,   48.23,   71.14   |
> |   FLTrust   |   60.47,   55.56,   58.02   |   83.68,   32.17,   57.93   |   94.55,   55.36,   74.96   |
> |  FoolsGold  |   66.38,   43.21,   54.79   |   86.10,   35.35,   60.73   |   94.02,   44.58,   69.30   |
> |    FLAME    |   60.30,   66.50,   63.40   |   83.48,   62.34,   72.91   |   92.85,   62.32,   77.59   |
> | Trim Median |   62.97,   57.44,   60.21   |   84.10,   34.23,   59.17   |   93.89,   55.23,   74.56   |
> |  Sageflow   |   65.98,   56.37,   61.18   |   87.99,   60.24,   74.12   |   93.81,   69.98,   81.90   |
> | **FedTGE**  | 63.69,   88.98,   **76.34** | 86.53,   71.29,   **78.91** | 94.40,   85.70,   **90.05** |
>
> After introducing a certain amount of noise, FedTGE still demonstrates significantly stronger defense performance compared to the baselines. We will also provide theoretical proofs to support this in the future.
>
> ---
>
> > `Question 2`: More non-IID conditions
>
> Thank you for your question. In the revised manuscript, we have provided more experimental reports under non-iid scenarios in Appendix B.
>
> **renyi** **(Non-iid-label-skew with alpha = 0.5 and a malicious proportion of 0.3)** :
>
> |   Methods   |     Cora  (A, R, V)     |    Pubmed  (A, R, V)    |   Physics  (A, R, V)    |
> | :---------: | :-------------------------: | :-------------------------: | :-------------------------: |
> |   FedAvg    |   61.72,   73.89,   67.80   |   85.88,   11.28,   48.58   |   94.34,   39.54,   69.93   |
> |   FLTrust   |   63.12,   59.66,   61.39   |  82.63,    8.98,    45.80   |   94.58,   31.25,   62.92   |
> |  FoolsGold  |   65.22,   63.98,   64.60   |   86.86,   10.91,   48.88   |   91.31,   61.05,   77.18   |
> |    FLAME    |   63.27,   70.08,   66.68   |   84.28,   14.43,   49.36   |   90.27,   42.64,   66.46   |
> | Trim Median |   60.90,   58.35,   59.62   |  86.39,    7.91,    47.15   |   94.24,   30.72,   62.48   |
> |  Sageflow   |   65.42,   44.91,   55.17   |  86.09,    0.35,    43.22   |   94.59,   26.51,   60.55   |
> | **FedTGE**  | 64.69,   88.91,   **76.60** | 86.98,   43.65,   **65.32** | 94.79,   86.79,   **90.79** |
>
> Some defense methods based on model updates lose most of their effectiveness in the label-skew scenario on the Pubmed dataset. In these scenarios, client data becomes biased toward specific labels, causing model updates to focus on majority class labels and making attacks harder to defend against. In contrast, FedTGE adjusts at the data level and does not rely directly on model updates, allowing it to maintain stronger defensive capabilities than the baselines.
>
> ---

---

> > ### Author Response · Authors · 2024-11-21
> > **[Part 2/2] Response to the Reviewer 8A86**
> >
> > > `Question 3`:  How does TESP prevent the incorrect inclusion of benign clients?
> >
> > Thank you for your detailed observation. We address your question from two perspectives: **(1)** **Benign clients are almost never excluded from the aggregation process.** The TESP component is designed to first calculate the energy distribution similarity between clients. A client is excluded from the aggregation process only if its similarity with all other clients falls below a specified threshold. Visualization results of the clustering process have be provided in Appendix D, where it can be clearly observed that no benign client, after the TEDC component adjusts energy, has a similarity distribution so low with all other clients that it would be excluded from aggregation. **(2)** **No benign client is assigned a very low aggregation weight, ensuring proper knowledge sharing.** The TESP clustering adjustment is based on the sum of the energy corresponding to each client, which is not directly related to similarity calculation. The smaller the sum of energy corresponding to a client, the larger the aggregation weight assigned to it. Therefore, the method ensures that benign clients are assigned appropriate aggregation weights.
> >
> > ---
> >
> > > `Question 4`: Complexity Analysis
> >
> > Thank you for this valuable suggestion. In the revised version of the manuscript, we have included a complete complexity analysis. The overall computational complexity of FedTGE can be formalized as:
> >
> > **TEDC**:
> > $$
> > O((3E \times F + 2D \times F + 2D + P_{BN}) \times EN)
> > $$
> > where $P_{BN}$ represents the parameters of the Batch Normalization (BN) layer in the model, $E$ denotes the number of edges in the dataset, $D$ and $F$ stand for the number of nodes and features respectively, and $EN$ refers to the epoch count used for energy calibration. In non-dense graphs, $E$ can be considered proportional to $D$, $P_{BN}$ can be neglected. Therefore, the formula can be further simplified as:
> > $$
> > O(E \times F \times EN)
> > $$
> > This indicates that the TEDC module has a linear relationship with the number of edges $E$ and is independent of the number of clients, enabling energy calibration and clustering for large-scale client numbers.
> >
> > **TESP**:
> > $$
> > O(E\times L+ N^2+K\times E + N\times M)
> > $$
> > We did not calculate the complexity of parameter aggregation because it is a process that exists in all FL systems. We only analyzed the complexity of the similarity propagation part. Here, $N$ represents the number of clients, $K$ and $L$ denote the number of propagation layers and the length of the energy distribution (i.e., the number of node), respectively. It is worth noting that the TESP module treats clients as nodes and the connections between clients as edges. Since $N$ and $E$ is usually a relatively small constant, the above formula can be further simplified as:
> >
> > $$
> > O(D+K)
> > $$
> >
> > This indicates that TESP module has a linear relationship with the number of nodes $D$, which demonstrates its suitability for large-scale datasets.
> >
> > ---

---

> > > ### Comment · Reviewer_8A86 · 2024-11-24
> > >
> > > I have carefully reviewed the rebuttal regarding Questions 3 and 4, while also considering the responses to Questions 1 and 2, and I believe the authors have effectively addressed all my concerns. I now consider this paper to meet the acceptance standards of ICLR and will accordingly increase my score.

---

> > > > ### Author Response · Authors · 2024-11-24
> > > >
> > > > ### Dear Reviewer 8A86,
> > > >
> > > > We are deeply grateful for your insightful comments and generous support of our research. Your constructive feedback on the scalability and adaptability of FedTGE has greatly enhanced the clarity and rigor of our work. It has been a privilege to address your concerns and refine our manuscript based on your suggestions. Once again, we sincerely thank you for your time and effort in reviewing our work.
> > > >
> > > > Best regards,
> > > >
> > > > Authors

---

> > ### Comment · Reviewer_8A86 · 2024-11-24
> >
> > I have carefully reviewed the authors’ rebuttal. The additional discussions and experiments for fairness of energy model non-IID settings appear convincing and adequately address my concerns regarding Questions 1 and 2. A minor suggestion is to include the discussion part for Question 1 in the paper if it has not already been added.

---

### Official Review · Reviewer_Z6ow · 2024-11-04

**Soundness:** 3
**Presentation:** 2
**Contribution:** 3
**Rating:** 6
**Confidence:** 4

**Summary:**

This paper addresses the challenges of backdoor attacks in Federated Graph Learning (FGL) by proposing an innovative defense method called FedTGE, which utilizes Topological Graph Energy. The method operates at both the client and server levels, injecting energy distribution knowledge into local models to differentiate between benign and malicious samples. It assigns low energy to benign samples while elevating the energy of constructed malicious substitutes, enabling the selection of trustworthy clients through clustering. At the server level, the energy elements uploaded by clients are treated as nodes in a global energy graph, facilitating energy propagation and enhancing the robustness of client selection. The experimental results validate the effectiveness of FedTGE under varying proportions of malicious clients and different data scenarios, demonstrating its capability to handle high data heterogeneity without requiring a validation dataset.

**Strengths:**

-	The introduction of the FedTGE framework is innovative, employing energy-based modeling to defend against backdoor attacks in federated graph learning (FGL).
-	The paper presents a comprehensive evaluation of the proposed method across various datasets and scenarios, demonstrating its effectiveness in both IID and non-IID settings.
-	The manuscript is well-structured, presenting a clear delineation of the proposed methodology, experimental setup, and results.

**Weaknesses:**

-	The efficacy of the FedTGE method heavily relies on the accurate modeling of energy distributions. To enhance the robustness of energy estimations, the authors should consider incorporating visual analysis of energy distributions prior to clustering.
-	The computational demands associated with energy graph construction and similarity propagation may hinder scalability. The authors should discuss about the computational overhead and the costs of federated transmission.
-	The authors should investigate the defense performance across different trigger structures and patterns to better understand and enhance the system's resilience under varied attack scenarios.
-	The method requires careful threshold selection for clustering energy elements, which could be subjective and impact performance. The authors should describe how to determine these thresholds.

**Questions:**

Refer to the weaknesses above.

---

> ### Author Response · Authors · 2024-11-21
> **[Part 1/1] Response to the Reviewer Z6ow**
>
> Dear Reviewer Z6ow
>
> We sincerely thank the reviewers for their valuable and constructive feedback, which has helped us identify ways to improve the clarity and quality of our manuscript. We have carefully addressed each comment, and our detailed responses are provided below.
>
> > `Weakness 1`: Visualization of the energy distribution
>
> To address your suggestion regarding the visualization of energy distribution during the clustering process, we have included kernel density estimation plots in Appendix E of the revised manuscript. These plots show the energy distribution before and after energy calibration for different trigger types. And the results demonstrate that our method effectively models node energy across various datasets and trigger types, clearly distinguishing between the energy distributions of malicious and benign clients.
>
> ---
>
> > `Weakness 2`: Complexity analysis
>
> Thank you for this valuable suggestion. In the revised version of the manuscript, we have included a complete complexity analysis in appendix. The computational complexity of FedTGE can be formalized as:
>
> **TEDC**:
> $$
> O((3E \\times F + 2D \\times F + 2D + P_{BN}) \\times EN)
> $$
> where $P_{BN}$ represents the parameters of the Batch Normalization (BN) layer in the model, $E$ denotes the number of edges in the dataset, $D$ and $F$ stand for the number of nodes and features respectively, and $EN$ refers to the epoch count used for energy calibration. In non-dense graphs, $E$ can be considered proportional to $D$, $P_{BN}$ can be neglected. Therefore, the formula can be further simplified as:
> $$
> O(E \\times F \\times EN)
> $$
> This indicates that the TEDC module has a linear relationship with the number of edges $E$, which demonstrates its scalability.
>
> **TESP**:
> $$
> O(E\\times L+ N^2+K\\times E + N\\times M)
> $$
> We only analyzed the complexity of the similarity propagation part. Here, $N$ represents the number of clients, $K$ and $L$ denote the number of propagation layers and the length of the energy distribution (i.e., the number of node), respectively. It is worth noting that the TESP module treats clients as nodes and the connections between clients as edges. Since $N$ and $E$ is usually a relatively small constant, the above formula can be further simplified as:
> $$
> O(D+K)
> $$
> This indicates that TESP module has a linear relationship with the number of nodes $D$, which demonstrates its suitability for large-scale datasets. For transmission costs, FedTGE does not introduce significant additional transmission pressure. Apart from transmitting the parameters of the backbone, FedTGE additionally transmits the energy distribution of each client to the server. Its length $L$ matches the number of nodes $D$ in the dataset, making it a completely acceptable transmission cost.
>
> ---
>
> > `Weakness 3`: More defense performance under various trigger types
>
> We agree that evaluating with additional trigger types would provide a more comprehensive assessment of the system's resilience. In the revised manuscript, we will provide experimental reports on more diverse trigger structures in Appendix B. The results show that FedTGE maintains higher defensive performance than the baseline across various trigger types. Below we present partial experimental results for the trigger type gta.
>
> **GTA (Non-iid-louvain with a malicious proportion of 0.3):**
>
> |   Methods   |     Cora  (A, R, V)     |    Pubmed  (A, R, V)    |   Physics  (A, R, V)    |
> | :---------: | :-------------------------: | :-------------------------: | :-------------------------: |
> |   FedAvg    |   78.54,   27.96,   53.25   |   86.86,   11.70,   49.28   |   93.42,   24.86,   59.14   |
> |   FLTrust   |   78.21,   43.01,   60.61   |   86.36,   36.05,   61.21   |   94.21,   59.26,   76.74   |
> |  FoolsGold  |   79.56,   25.60,   52.58   |   89.15,   16.70,   52.87   |   94.68,   40.56,   67.62   |
> | Trim Median |   79.49,   33.06,   56.26   |   86.63,   14.58,   50.60   |   93.55,   27.39,   60.47   |
> |  Sageflow   |   79.95,   29.49,   54.72   |   89.04,   16.70,   52.87   |   94.54,   54.69,   74.62   |
> | **FedTGE**  | 80.23,   75.23,   **77.73** | 87.56,   55.34,   **71.45** | 94.39,   72.37,   **83.38** |
>
> ---
>
> > `Weakness 4`: Selection of the threshold
>
> FedTGE utilizes the advanced FINCH technique for clustering the energy distribution of clients, which typically does not require pre-setting a threshold. However, in our TESP component, a threshold is applied to determine whether the energy similarity between two clients is sufficient for energy propagation. This threshold is designed to **enhance the robustness** of FedTGE. as shown in Appendix D. Even if a malicious client is mistakenly selected, it is difficult to maintain a high energy distribution similarity with benign clients, thus further helping to exclude malicious clients. Therefore, it is typically set to a relatively high value, such as the default value of 0.8 in the paper, which has been shown to achieve effective defense.

---

> > ### Comment · Reviewer_Z6ow · 2024-11-21
> > **Thanks for the response**
> >
> > I read this rebuttal and appreciate for authors' response. Introduced engergy models into backdoor defence is interesting and the experiments show its feasibility. Minor suggestion is that the application scope and depoyment details can be provided more.

---

> ### Author Response · Authors · 2024-11-21
> **Thank you for your response to our rebuttal**
>
> ### Dear Reviewer Z6ow,
>
> Thank you for your kind feedback on our method and for recognizing the novelty of using energy models in backdoor defense.
>
> **Application Scope**: Our method is effective for the current scenarios and highly extensible to graph-level tasks, making it applicable across a wide range of use cases. Furthermore, our approach demonstrates strong scalability, ensuring its feasibility in handling larger datasets and federated scenarios with more clients.
>
> **Deployment Details**: We have provided implementation details in the manuscript (Sec 5.1) and anonymous code to facilitate understanding and reproducibility.  The experiments use NVIDIA GeForce RTX 3090 GPUs as the hardware platform, coupled with Intel(R) Xeon(R) Gold 6240 CPU @ 2.60GHz.  Should you have any specific aspects of the deployment process in mind that require further elaboration, we would be more than happy to clarify or expand upon them.
>
> If you have any additional questions or suggestions, please let us know. We are committed to addressing all concerns to improve our work further.
>
> Thank you again for your thoughtful review and support. We hope this will contribute to a higher evaluation of our submission.

---

### Meta-Review · Area_Chair_SSLn · 2024-12-21

**Metareview:**

This paper tackles the challenge of backdoor attacks in Federated Graph Learning and also introduces a defense strategy based on Topological Graph Energy. The proposed approach incorporates structural distributional knowledge into the model. At the client level, it uses energy modeling to distinguish between benign and malicious samples, while at the server level, it creates a global energy graph for energy propagation, effectively detecting and filtering out malicious clients. Experimental results also demonstrate FedTGE's effectiveness.

Strength:

1. The proposed methods are interesting and novel.

2. The paper's structure and writing is good.

Weakness:

Although the authors discussed their method's complexity and claimed they can be extended to large-scale datasets, no further experiments for it to demonstrate the proposed methods effectiveness on large-scale datasets.

In summary, this paper is a good paper with a novel method design, comprehensive evaluations and good writing. All the reviewers show positive attitude towards this paper. And I also suggest to accept it as a spotlight.

**Additional Comments On Reviewer Discussion:**

The reviewers and authors discuss the papers' additional evaluations, scenarios, complexities, etc. These discussions also improve this paper.

---

### Decision · Program_Chairs · 2025-01-22

Accept (Oral)